# Impact of karst areas on runoff generation, lateral flow and interbasin groundwater flow at the storm-event timescale

Martin Le Mesnil[1,2,3], Roger Moussa[1], Jean-Baptiste Charlier[2,3], Yvan Caballero[2,3]

[1]LISAH, Univ. Montpellier, INRAE, IRD, Institut Agro, Montpellier, France

[2]BRGM, Univ. Montpellier, Montpellier, France

[3]G-eau, INRAE, CIRAD, IRD, AgroParisTech, Institut Agro, BRGM, Montpellier, France

Correspondence to: Martin Le Mesnil (martin.le-mesnil@inrae.fr)

**Abstract.** Karst development influences the hydrological response of catchments. However, such impact is poorly documented
and even less quantified, especially over short space and time scales. The aim of this article is thus to define karst influence on the different hydrological processes driving runoff generation, including interbasin groundwater flow (IGF) for elementary catchments at the storm-event time scale. IGF are estimated at the scale of the river reach, by comparing inlet and outlet flows, as well as the effective rainfall from the topographic elementary catchment. Three types of storm-event descriptors (characterising water balance, hydrograph shape and lateral exchanges) were calculated for the 20 most important storm events
of 108 stations in three French regions (Cévennes Mountains, Jura Mountains and Normandy), representative of different karst settings. These descriptors were compared and analysed according to catchment geology (karst, non-karst, or mixed) and seasonality in order to explore the specific impact of karst areas on water balance, hydrograph shape, lateral exchanges and hydrogeological basin area. A statistical approach showed that, despite the variations with study areas, karst promotes: i) Higher water infiltration from rivers during storm events; ii) Increased characteristic flood times and peak-flow attenuation;
and iii) Lateral outflow. These influences are interpreted as mainly due to IGF loss that can be significant at the storm-event scale, representing around 50% of discharge and 20% of rainfall in the intermediate catchment. The spatial variability of such effects is also linked to contrasting lithology and karst occurrence. Our work thus provides a generic framework for assessing karst impact on the hydrological response of catchments to storm events; moreover, it can analyse flood-event characteristics in various hydro-climatic settings, and can help testing the influence of other physiographic parameters on runoff generation.

## 1      Introduction

Understanding runoff generation requires a good knowledge of the different processes involved in catchment response to rainfall events, i.e. how precipitation is converted into underground, subsurface or surface flow. These processes are affected by several factors, such as the thickness and nature of soil, land use, hydrological conditions, or geology. While most of these can be documented or measured, it may be difficult to define the role of geology in a comprehensive way, especially when
underground drainage networks are involved, as in karst areas. Karstification, the result of carbonate-rock dissolution, promotes infiltration and groundwater flow through enlarged fissures and voids (Bakalowicz, 2005), locally dramatically reducing drainage density on the surface and thus affecting the hydrological response of a catchment. These groundwater flows, if they do not return to surface within the considered catchment, can be considered as interbasin groundwater flows (IGF). IGF amplitude and sign can be linked to the area of the hydrogeological basin, which can be different from the area of
the topographic one.

Karst impacts on flood processes are mostly documented through case studies. As an example, Zanon et al. (2010) showed that during a flash flood in 2007 in Slovenia, karst area reduced flooding, which was more important in a non-karst neighbouring zone, receiving less precipitation. Likewise, Delrieu et al. (2005) observed, for an exceptional storm event in 2002, lower runoff coefficient values for the karstic catchment compared to the hard-rock catchment in the eastern zone of the
Cévennes Mountains. De Waele et al. (2010) and Charlier et al. (2015, 2019) determined that, depending on the location on

the river profile, karst areas could result in streamflow losses or gains due to the high spatial variability of the hydrogeological karst features. Other frequently described processes are groundwater rising leading to reduced infiltration and important surface runoff (López-Chicano et al., 2002; Bonacci et al., 2006), and backflooding/sinkhole flooding due to conduit constriction (Maréchal et al., 2008; Bailly-Comte et al., 2009).

The diversity of observed processes during storm event in karst catchments does not allow drawing a straightforward analysis on the control of karst in flood runoff generation. In the purpose of understanding the mechanisms involved in this control, there is a need for regionalized studies, covering a large-scale analysis of karst impact over short time periods when the catchment reacts after storm events. It is reasonable to think that karst can alternately increase or decrease storm impacts, depending on its capacity to infiltrate precipitation or to release stored water, i.e. depending on the direction of IGF it promotes.

Despite the early conceptualization of IGF (Eakin, 1966), its major role in karst hydrological processes is tackled by very few studies (Le Moine et al., 2007; Lebecherel et al., 2013). Some authors tried to improve model capacities to reproduce karst-based IGF, such as Nguyen et al. (2020) with SWAT, Le Moine et al. (2008) with GR4J or Scanlon et al. (2003) comparing a distributed and a lumped model. Nevertheless, those studies dedicated to the improvement of model performance are not devoted to describe and understand all flood components in karst catchments.

On one hand, most studies including karst system descriptors are based on a purely hydrogeological point of view, and are very integrative, as they tend to characterize karst aquifer as a whole, by analysing daily spring discharge. Gárfias-Soliz et al. (2010) fond that system memory, response time and mean input-output delay are relevant indicators for karstification, in addition to a necessary consideration of the structural complexity and heterogeneity of the lithology. Hartmann et al. (2013), using 10 system signatures, performed a model parameters sensitivity analysis to investigate their links with hydrological

processes on five Europe and Middle East karst sites. Basha et al. (2020) proposed six recession curve equations for the classification of karst aquifers, depending on their flow characteristics. On the other hand, some studies accounting for a spatialization of catchments focus on low-flow issues and surface-water/groundwater interaction (e.g., Covino et al., 2011, Mallard et al., 2014). Moreover, most regionalization works tend to spatialize annual indices (Sivapalan et al., 2011) or model parameters (Parajka et al., 2005; Oudin et al., 2008), and usually exclude catchments with identified IGF (Merz and Blöschl,

2004) or karst areas (e.g., Laaha and Blöschl, 2006).

Regional spatial analyses need to be based on reliable data at the highest resolution available. For this purpose, the scale of the elementary catchment - i.e. subdivision of a basin following available gauging stations – appears to be the best resolution for long-term monitoring. Elementary catchment can be either the drained area of a headwater catchment controlled by a gauging station, or the drained area between two gauging stations (intermediate catchment). When considering surface and groundwater

components, the delineation method of elementary catchments is questionable (topographic vs. hydrogeological boundaries). Despite the importance of groundwater processes in karst areas, topographic catchment delineation remains a more robust reference, for several methodological reasons. First, IGF can be defined as groundwater flow crossing topographic divides, as this concept emerged with the evidence of certain groundwater systems extending beyond the limits of valleys (Eakin, 1966). A perfectly delineated groundwater basin would then show IGF equal to zero. For this reason, studies related to IGF often use

the topographic catchment spatial reference (Genereux et al., 2005; Schaller and Fan, 2009; Bouaziz et al., 2018; Nguyen et al., 2020; see also a synthesis in Fan, 2019). Second, although in karst catchments groundwater contributes to flood flow, surface runoff has to be considered obviously as an important component of this latter. Exclusive consideration of hydrogeological catchments could thus lead to wrong surface contribution assessment depending on their surface drainage network. Third, as some groundwater flows are aligned with the main surface drainage axis, hydrogeological catchments would

encompass the whole river, making it impossible to study the spatial variability of parameters along the river, at the elementary catchment scale. Finally, topographic delineation is reliable and easily reproducible, while groundwater delineation is characterized by a strong uncertainty and variability in karst areas.

This article aims at providing a new methodology to characterize the spatial variability of karst influence on hydrological processes affecting runoff generation, including IGF, at the storm-event time scale. IGF are estimated at the scale of the river reach, by comparing inlet and outlet flows, as well as the effective rainfall from the topographic elementary catchment. The present study complements the previous work by Le Mesnil et al. (2020), which described the role of karst areas using annual water-budget indicators at the elementary catchment scale. Here, descriptors are calculated for major storm events at 108 stations in three areas in France (Cévennes Mountains, Jura Mountains and Normandy) with different karst settings. The descriptors are of three types: water balance, hydrograph shape and lateral exchange. Water-balance descriptors are obtained from discharge and precipitation depths, analysing the respective importance of the different flows during storm events. They help understanding how catchments transform precipitation into surface- and underground flow. Such descriptors are of a great interest to assess the spatial variability of catchment hydrological response (Sivapalan et al., 2011). Hydrograph-shape descriptors are derived from catchment hydrographs recorded at inlet and outlet stations during storm events. They describe hydrological processes (Raghunath, 2006) and, when analysed on successive stations, help characterizing flood wave routing. Lateral-exchange descriptors are based on lateral hydrographs, simulated with the diffusive wave equation (DWE, Moussa, 1996) applied between two gauging stations (for intermediate catchments only). They provide information on lateral inflow and outflow at the elementary catchment scale. Lateral flow is mainly the result of IGF, effective rainfall, variations in aquifer storage and overbank phenomena. A particular analysis is performed on IGF, which are expressed in depth as well as in fictive catchment area variation, and of which seasonal influence is discussed.

The three types of descriptors are compared and analysed according to the catchment geology type (classified as 'karst', 'non-karst', or 'mixed'), in order to explore its impact on runoff generation processes. The paper thus provides a framework for assessing the impact of a given physiographic parameter on the hydrological response of catchments to storm events.

## 2    Methodology

### 2.1    General methodology

We calculated 15 descriptors (five for each of the three types) of catchment response to storm events and assessed their variability as a function of karst occurrence. To this end, we grouped elementary catchments into three different geology types, based on relative areas of their main geological formations. Catchments underlain by only karstified rock are in the karst group (K), whereas catchments containing only non-karstified rock are in the non-karst group (NK). Any catchment with a combination of both karstified (>10%) and non-karstified rocks is in the mixed group (M). The karstifiable nature of rocks was assessed with the BDLISA database (Sect. 3.1.2). Descriptors were calculated for the 20 strongest storm events for all catchments (Sect. 2.2). The values obtained for each group (K, M, NK) were compared with each other to assess karst influence on them. Then, for K catchments only, the three study areas (Cévennes, C; Jura, J; Normandy, N, see Sect. 3.2 for area descriptions) were compared to assess the area-specific nature of karst influence. Our framework was kept as generic as possible, to propose an approach that would be easily adaptable to another investigation field. Consequently, the karst-catchment classification can be replaced by any other physiographical typology.

Descriptors are complementary but not necessarily independent from each other. They are of three types, and were chosen to provide relevant information on different processes:

- Water-balance descriptors show the respective importance of the different flows occurring during storm events and allow understanding how catchments transform rainfall into surface- and underground flow. Such descriptors are volumes (expressed as depth [mm]), or volume ratios. They can be calculated at headwater-catchment or intermediate-catchment outlets, the latter involving subtracting inlet flow from outlet flow. This was applied to the 108 elementary catchments.

- Hydrograph-shape descriptors describe the dynamics of storm events and flood-wave routing. They combine peakflow variation, characteristic times and flood-wave celerity. Characteristic times can be obtained from any measured hydrograph, whereas peakflow variation and celerity are evaluated between inlet and outlet hydrographs (for intermediate catchments only), considering catchments with only one inlet station; some intermediate catchments having several inlets when covering the confluence of streams. This was applied to all 108 elementary catchments for characteristic times and to the 36 intermediate catchments with only one inlet regarding peakflow variation and celerity.

- Lateral-exchange descriptors provide information on the dynamics of lateral inflow and outflow affecting an elementary catchment reach, as well as on the respective contributions of channel diffusivity and lateral exchanges to peakflow variations. This analysis is based on lateral hydrographs, simulated with the DWE, applied to intermediate catchments with one inlet. Lateral exchanges may be a combination of effective rainfall ($P_{eff}$) over the elementary catchment, IGF, aquifer storage variation and overbank phenomena. Overbank flow is considered as a minor process assuming that the overflow water returns to the river after a relatively short time during the recession. Thus, a water balance including $P_{eff}$ allows discussing the importance of IGF, along with aquifer storage variation. This was applied to the 36 intermediate catchments with only one inlet for lateral exchange dynamics, and to all 108 elementary catchments for IGF assessment.

## 2.2    Flow assessment at the elementary catchment scale

The 15 descriptors were calculated on elementary catchments for 20 selected strongest storm events. For the sake of representativeness, this selection was based on both rainfall and streamflow records. From the available data time series spanning several decades (see 3.1 for more details), the ten strongest precipitation and ten strongest streamflow events were extracted. Care was taken not to select the same events via the two extraction methods.

Streamflow is measured at several gauging stations along a given river, defining elementary catchments. For the most upstream station of a river, the elementary catchment corresponds to the ordinary topographic catchment. Otherwise, the elementary catchment is an intermediate one, covering the portion of the basin drained between two gauging stations (Fig. 1). In the case of intermediate catchments, streamflow is calculated following Eq. (1) as the difference between outlet flow ($Q_O$) and inlet flow ($Q_I$). A hydrograph decomposition is operated for separating the quick (stormflow $Q_S$) and slow (underground flow $Q_U$) flow components, as shown in Eq. (2) and (3) (see Appendix A for more details). The $Q_S$ and $Q_U$ components of the intermediate catchment are also obtained from the difference between inlet and outlet flow components as shown in Eq. (4) and (5). For these intermediate catchments, the $Q$, $Q_S$ and $Q_U$ values can thus be negative; note also that the associated uncertainties are twice higher than those of measured flows ($Q_O$ and $Q_I$).

$$Q = Q_O - Q_I \tag{1}$$
$$Q_I = Q_{I,U} + Q_{I,s} \tag{2}$$
$$Q_O = Q_{O,U} + Q_{O,s} \tag{3}$$
$$Q_S = Q_{O,s} - Q_{I,s} \tag{4}$$
$$Q_U = Q_{O,U} - Q_{I,U} \tag{5}$$

In addition to inlet and outlet flows, a lateral flow ($Q_L$) is calculated. This flow is mainly a combination of IGF, effective rainfall ($P_{eff}$) over the elementary catchment, aquifer storage variation and overbank phenomena (Eq. 6). Overbank phenomena is neglected here (see Sect. 2.1). The $Q_L$ hydrograph is simulated using an inverse model, solving the DWE accounting for lateral flow between inlet and outlet stations, assuming a uniform lateral distribution of exchanges along the river reach (see Appendix B for more details).

$$Q_L = IGF + Peff + \delta \tag{6}$$

Finally, after an estimation of $P_{eff}$, (Appendix C), all measurable inflow ($Q_I$ and $P_{eff}$) and outflow ($Q_O$) are known. Hence, IGF

(groundwater flowing inside or outside the elementary catchments delimited by topographic boundaries) can be calculated for each storm event, along with the aquifer storage (noted $\delta$). Figure 1 shows all considered flows and their corresponding hydrographs, at the spatial scale of an elementary catchment and the time scale of a storm event.

## 2.3 Descriptors

### 2.3.1 Water balance descriptors

For each storm event, volumes of the different flows in an elementary catchment are calculated (Fig. 1). Total discharge volume noted $V_Q$, is calculated as:

$$V_Q = V_O - V_I = (V_{O,S} + V_{O,U}) - (V_{I,S} + V_{I,U}) \tag{7}$$

with $V_I$, $V_O$ being the volumes of total streamflow (expressed as water depths in mm) and $V_{I,S}$, $V_{I,U}$, $V_{O,S}$, $V_{O,U}$ those of the quick- and slow-flow components of inlet $Q_I$ and outlet $Q_O$, respectively. $V_P$ is the volume of precipitation falling on the

elementary catchment during the storm event. The values being dependent on the catchment surface, they are normalized by the topographic area (A) of the considered catchment and expressed in water depth [mm]. From these volumes, five water-balance descriptors are calculated (see Appendix D for equations):

- $V_S$: event stormflow [mm];
- $V_U$: event baseflow [mm];

- $R_C$: event runoff coefficient [dimensionless], calculated as total event streamflow divided by event rainfall;
- $R_{C,S}$: event storm runoff coefficient [dimensionless], calculated as event stormflow divided by event rainfall;
- $S_C$: event stormflow coefficient [dimensionless], calculated as event stormflow divided by total event streamflow.

### 2.3.2 Hydrograph-shape descriptors

For each storm event, five more descriptors were calculated, characterizing hydrograph morphology and storm-event dynamics. Characteristic times are of a great interest in storm hydrology and constitute a widely used framework, allowing convenient catchment and event comparisons (Bell and Om Kar, 1969). Here, we use three of them. The time constant of the rising limb $T_{Ri}$ corresponds to the time duration needed for streamflow to increase from half-peakflow ($0.5\ Q_{O,X}$) to peakflow ($Q_{O,X}$). $T_{08}$ is the time duration for $Q_O(t) > 0.8\ Q_{O,X}$, important in terms of operational flood management. $T_{Re}$ is the

characteristic recession time, obtained from the linearization method of Maillet (1905), which approximates the recession curve by an exponential function as shown in Eq. (8).

$$Q_{Maillet}(t) = q \cdot e^{-\beta t} = q \cdot e^{-t/T_{Re}} \tag{8}$$

with $\beta$ the recession coefficient and q the discharge at the beginning of the recession phase. The $T_{Re}$ time constant is the duration needed for streamflow to decrease by a factor $e \approx 2.7$ during recession.

In the case of intermediate catchments, the peakflow variation ($\Delta$) is calculated as the difference between outlet station peakflow ($Q_{O,X}$) and inlet station peakflow ($Q_{I,X}$), normalized by rainfall. A parameter $C_G$ is also calculated, used as the celerity for applying the DWE (Appendix B). $C_G$ is equal to the river reach length (l) divided by the elapsed time ($T_G$) between QI and QO gravity centres ($G_I$ and $G_O$). The five hydrograph-shape descriptors (see Appendix D for formulae) are:

- $\Delta/V_P$, peakflow variation normalized by rainfall [$s^{-1}$];

- $T_{Ri}$, event time duration of rise [h];
- $T_{08}$, event time duration to 80% of peakflow [h];
- $T_{Re}$, event characteristic recession time as per Maillet (1905) approximation [h];

- $C_G$, celerity based on elapsed time between hydrograph gravity centres [m.s$^{-1}$].

### 2.3.3 Lateral-exchange descriptors

For catchments with one inlet station, a lateral-flow hydrograph is simulated, using the solution of the inverse problem of the DWE assuming that the lateral flow is uniformly distributed along the channel (Moussa, 1996; see Appendix B). The DWE equation has two free parameters, the celerity C (m.s$^{-1}$) and the diffusivity D (m².s$^{-1}$) of the flood wave. First, $Q_I$ is routed using the DWE without lateral flow for given values of C and D, simulating a theoretical outflow without lateral exchange, noted $Q_{I,R}$ on Fig. 3. The inverse problem supposes known $Q_I$, $Q_O$, C and D values, simulating the lateral flow $Q_L$. Sensitivity analysis

of the DWE to the two parameters is largely available in the literature, showing that the DWE is more sensitive to parameter C than D (Moussa and Bocquillon, 1996; Cholet al., 2017; Charlier et al., 2019). Here, C was assumed to be equal to $C_G$ and we used a fixed value for D = 10,000 m²/s (for medium size catchments of 100 to 500 km²; Moussa and Bocquillon, 1996; Todini, 1996) in a matter of parsimony, which is acceptable as the model is much more sensitive to C than to D. The DWE solution was validated experimentally under controlled conditions (Moussa and Majdalani, 2019), and has also been

implemented on natural karst catchments (Charlier et al., 2015, 2019; Cholet et al., 2017).

From the simulated lateral hydrograph, five descriptors were calculated. First, the peakflow-variation descriptor Δ was decomposed into two components $\Delta_D$ and $\Delta_L$, the respective contributions of channel diffusivity and lateral exchanges (Charlier et al., 2019). $\Delta_D$ and $\Delta_L$ are also normalized by $V_P$. In the case of a reach without lateral exchanges, Δ is negative and only due to channel diffusivity (equal to $\Delta_D$, Fig. 3(a)). In the case of a gaining catchment, Δ can be positive or negative, with

compensating contributions of channel diffusivity and lateral inflow (Fig. 3(b)). In the case of a losing catchment, Δ is negative, with cumulated contributions of channel diffusivity and lateral outflow (Fig. 3(c)). Second, volumes of lateral inflow and outflow ($V_{L+}$ and $V_{L-}$) were calculated from the lateral hydrograph and normalized by event rainfall. Note that $Q_L$ can be successively positive and negative within a single storm event (Fig. 1).

As simulated lateral hydrographs are based on the difference between discharge volumes at the inlet and outlet of catchments,

they also integrate precipitation on the elementary catchment during storm events. In order to focus on groundwater exchanges and suppress the influence of effective rainfall ($P_{eff}$), this latter has to be estimated. It was done using three different modelling approaches, providing a range for this component characterized by major uncertainties. The three approaches, respectively based on the methods of Thornthwaite (1948), Dingman (2002), and the GR model (Edijatno et al., 1999), are described in Appendix C. The final value is the mean of those obtained from the three methods.

The remaining lateral-exchange term combines IGF and potential aquifer storage variation (δ). δ can either be positive, corresponding to an aquifer recharge, or negative, corresponding to an aquifer draining. Aquifer draining is less likely during storm events, as important rainfalls generally occur. In case of an aquifer recharge, and as our analysis is performed on the whole storm-event period (including the entire recession), a substantial part of the infiltrated water should be released, either inside the considered elementary catchment or outside of it. In the first case, the considered amount of water is accounted for

in the $V_O$ term and will not influence the lateral-exchange term. In the second case, the released water actually constitutes an IGF. For these reasons, we use the term IGF* for the obtained water-balance term. For each storm event, IGF*, combining IGF and potential aquifer storage variation (δ), is calculated as in Eq. (9):

$$IGF^* = IGF + \delta = V_Q - V_{Peff} = V_O - V_I - V_{Peff} \tag{9}$$

The five lateral exchange descriptors (see Appendix D for formulae) are:

- $\Delta_D/V_P$, part of peakflow variation due to channel diffusivity, normalized by rainfall [s$^{-1}$];
- $\Delta_L/V_P$, part of peakflow variation due to lateral exchanges, normalized by rainfall [s$^{-1}$];
- $V_{L+}/V_P$, event lateral inflow, normalized by rainfall [dimensionless];
- $V_{L-}/V_P$, event lateral outflow, normalized by rainfall [dimensionless];

- IGF*/$V_P$, event interbasin groundwater flow including potential aquifer storage variation (δ), normalized by rainfall [dimensionless].

## 2.4 Statistical approach

Once all descriptors have been calculated, a statistical analysis is performed for comparative purposes. For each descriptor, the obtained values are grouped in different samples, by i) geology type (K, M, NK), and ii) study area (C, J, N) for karst catchments only (K geology type). This allows characterizing the impact of karst areas on the hydrological response and provides additional information on regional specifics of this impact.

The results are presented as boxplots, to discuss how the distribution of the descriptors varies for all samples. Then, statistical tests assess the significance of the results. Twin-sample t-tests are performed successively on K *vs*. NK, M *vs*. NK and M *vs*. K catchments, and—only for K catchments—on C *vs*. J, C *vs*. N and J *vs*. N. Since it is not assumed that the two data samples are from populations with equal variances, the test statistics under the null hypothesis have an approximate Student's 't' distribution with a number of degrees of freedom given by Satterthwaite's approximation (Satterthwaite, 1946). This test provides a decision for the null hypothesis that the data in paired tested samples come from independent random samples from normal distributions with equal means, and equal but unknown variances. The result is 1 if the test rejects the null hypothesis at the 5% significance level, and 0 otherwise.

## 3 Data sets and study areas

### 3.1 Data sets

#### 3.1.1 Temporal data

Temporal data used in this paper are:
- Hourly streamflow data from the French public streamflow database 'Banque Hydro' (http://www.hydro.eaufrance.fr/), managed by the French Regional Environment Directions (*Direction Régionale de l'Environnement, de l'Aménagement et du Logement, DREAL*). These data are produced by interpolation of water-depth measurements at a variable infra-hourly time step and converted into streamflow values.
- Hourly rainfall data are from Comephore (Tabary, 2007; Tabary et al., 2007) covering 1997 to 2005 and Antilope (Champeaux et al., 2011) from 2006 to 2018. Both datasets are measurement reanalyses edited by the French public meteorological service Météo France (http://www.meteofrance.fr/). They have a 1-km² spatial resolution and consist in radar rainfall measurements, calibrated to fit the data from surface precipitation gauges.
- Daily potential evapotranspiration depths are from "Safran" (*Système d'Analyse Fournissant des Renseignements Atmosphériques à la Neige*; Vidal et al., 2010), edited by the French meteorological service (Météo France). They are used for estimating effective rainfall (Appendix C).

Hourly rainfall data are available from 1997 onward, and hourly streamflow data periods vary depending on the catchments. Table 1 shows periods of availability of both data sets for 11 hydrographic catchments in three areas of France, covering a total area of almost 25,000 km². Attention was paid to use only validated streamflow data from gauging stations with a hydrological significance, i.e. not influenced by human activities such as damming or pumping. Nevertheless, a 10% uncertainty is associated with streamflow data, which can be higher during extreme storm events due to uncertain rating equations or measuring ranges.

### 3.1.2 Spatial data

Spatial data used in this paper are:

- Boundaries of topographic catchments, from the French National Watershed Database (*Base Nationale des Bassins Versants*, BNBV). It was edited by the French Central Service for Hydrometeorology and Support on Flood Forecasting (*Service Central d'Hydrométéorologie et d'Appui à la Prévision des Inondations*, SCHAPI) and the French Research Institute in for Agriculture, Food supply and the Environment (*Institut National de Recherche en Agriculture, Alimentation et Environnement* INRAE).
- BDLISA database (https://bdlisa.eaufrance.fr/), which describes the properties of hydrogeological entities and aquifers in France, edited by the French Geological Survey (*Bureau de Recherches Géologiques et Minières*, BRGM).
- Map of available soil-water capacity from INRAE (Le Bas, 2018), used for estimating effective rainfall (Appendix C).

### 3.2 Study areas

### 3.2.1 General settings

The previously described methodology was applied to three areas in France, including 11 hydrographic basins and representing a total area of 25,000 km². The three areas are totally or partially karstified, with different geological and hydrometeorological settings. From south to north, they belong to the regions of Cévennes Mountains, the Jura Mountains and Normandy. Figure 4 shows the locations of the 11 hydrographic basins, on the map of French karst aquifers (4A), and the gauging-station networks with geology type of the elementary catchments on maps 4B, C and D.

In the Cévennes Mountains (Fig. 4(b)), six hydrographic basins were studied, including 51 gauging stations. They are mostly so-called binary karst basins, with head catchments on hard rock receiving around 1500 mm/year precipitation, and median and downstream parts underlain by limestone plateaus, with around 1000 mm/year rainfall.

The Jura Mountains region corresponds to the Doubs River basin, a few kilometres upstream from its confluence with the Saône River, which includes 39 gauging stations. Outcrops mostly consist of extensively karstified Jurassic limestone and marl, except in the extreme northern and western parts of the region. Precipitation follows a strong elevation gradient, with annual values ranging from 1700 mm on upstream catchments at heights of 1400 m (a.s.l), to 1200 mm at the outlet at an elevation of around 200 m (a.s.l).

In Normandy, four hydrographic basins covered 18 gauging stations. The two eastern basins are tributaries of the Seine, the other two being coastal basins. The climate is maritime and annual rainfall ranges from 700 to 1000 mm. Rivers in the eastern part of the area drain chalky limestone with karst covered by clay. The mid-western zone is underlain by Jurassic limestone, corresponding to the western border of the Paris Basin.

### 3.2.2 Hydrometeorology and physiography of study areas

The surface area of elementary catchments depends upon the location of gauging stations. They have similar ranges for each geology type and study areas (Fig. 5(a)), and the potential bias induced by scale effect is limited. Figure 5(b) shows the strong variability of reach mean slope with geology type. This can be explained by the morphology of karst plateaus that are prone to intense erosion, forming canyons with low slopes controlled by the base level. Moreover, higher-elevation ground is mostly underlain by hard-rock (i.e. non-karst) terrains. Contrasting slopes should thus not be seen as a bias, but as the result of an intrinsic characteristic of limestone areas, coinciding with karst occurrence. Reach mean slope on karst catchments has similar values for the three study areas, with a maximum variation in median slope of 5‰ from Normandy to the Cévennes.

Figures 5(c) and 5(d) present the distribution of precipitation and discharge depth for the 20 strongest storm events (see Sect. 2.1) in the 108 elementary catchments, grouped by geology type (K: karst, M: mixed, NK: non-karst), and by study area (C: Cévennes, J: Jura, N: Normandy) for K catchments. A major contrast in precipitation depth is highlighted, with a median value of 100 mm per event for the Cévennes catchments, which is higher than the maximum recorded value for the Normandy catchments (80 mm). Jura Mountains catchments have an intermediate position, with a median rainfall-event depth of around 50 mm. A similar variation is observed for geology types, median-event rainfall depth increasing from 45 mm on K catchments to 105 mm on NK catchments. This is partly due to the Cévennes upstream catchments of non-karst hard rock receiving intense rainfall. Nevertheless, this has no major influence on the descriptor values since they are normalized by rainfall. The same trend is seen for flow depths, with median values of 20, 10 and 2 mm and 50, 15 and 5 mm for C, J, N and NK, M, K catchments, respectively.

## 4 Results

### 4.1 Water balance descriptors

The distribution of water-balance descriptors is presented on Fig. 6 (top row) i) by geology type (K: karst, M: mixed, NK: non-karst), and ii) by location for karst catchments only (C: Cévennes, J: Jura, N: Normandy). Stormflow and baseflow depth ($V_S$ and $V_U$) are influenced by geology type, both decreasing by a factor 10 from NK to K reaches. $V_P$ may have an influence on these results, but cannot explain all of this correlation since it only varies by a factor 2. $V_S$ and $V_U$ being calculated from differences between inlet and outlet runoff (see Appendix D), show that karstified catchments generally produce less streamflow than other ones, regarding both quick- and slow-flow components. K and M catchments have their first $V_S$ and $V_U$ quartiles close to zero, highlighting the major streamflow losses along their reaches. $V_S$ and $V_U$ values also vary with location, $V_S$ marking the typical rainfall intensity of each climatic region (Cévennes, Jura and Normandy in descending order; Fig. 5(a)), whereas $V_U$ seems to be more influenced by baseflow index information that is lower in Cévennes, probably due to a higher number of NK catchments.

Runoff coefficients are also significantly influenced by geology type, $R_C$ values dropping from 0.42 on NK to 0.14 and 0.19 on K and M, respectively. This trend is observed in a comparable way in the three areas, all showing $R_C$ values between 0.1 and 0.2 for their K reaches. Storm-runoff coefficients show a similar behaviour, with even more contrasting values between K and NK reaches ($R_{C,S}$ being five-fold higher in NK, against three-fold for $R_C$). Stormflow coefficient $S_C$ corresponds to the part of streamflow due to quick runoff. It decreases from 0.45 to 0.3 in K reaches and also has a geographic pattern, with Cévennes K catchments having 50% of quick-flow components, against 25% for Jura and Normandy K catchments. This correlates with lower $V_U$ values in the Cévennes K catchments, even with higher rainfall depths.

Water-balance descriptors globally show that karst areas promote more infiltration (lower $R_C$), the infiltrated water not being released through streamflow at the storm-event timescale (lower $S_C$).

### 4.2 Hydrograph-shape descriptors

Distributions of hydrograph-shape descriptors are shown on Fig. 6 (middle row) i) by geology type, and ii) by study area for karst catchments only. Median values of peakflow variation normalized by rainfall ($\Delta/V_P$) vary from 4.6 E10$^{-6}$ s$^{-1}$ for NK catchments to 1.9 E10$^{-6}$ and 0.5 E10$^{-6}$ m$^3$.s$^{-1}$.km$^{-2}$ for M and K catchments, respectively. It shows that the larger the karst outcrops, the smaller is peakflow amplification. This trend was noted in all study areas, even though Cévennes K catchments show more peakflow amplification than others (and especially a large $\Delta$ variability).

Karst catchments have a median rise duration $T_{Ri}$ of 11 hours, whereas M and NK rises take 6 and 4 hours, respectively. Streamflow increase is thus considerably slower in karst reaches. Median durations at 80% of peakflow $T_{08}$ were also correlated

with geology type, with values of 11, 6 and 3 hours for K, M and NK catchments respectively, showing that the slower discharge increase is associated with a buffered peakflow. The median value of the recession time constant ($T_{Re}$) is 44 hours for K catchments, and 28 hours for M and NK ones. Typical recessions in karst reaches are thus 50% slower than in others. The characteristic times $T_{Ri}$, $T_{08}$ and $T_{Re}$ are influenced by karst in different ways depending on study area. For each of them, Cévennes catchments show shorter durations, followed by Normandy with durations close to the median values of the whole

K-catchments sample, and by the Jura.

The flood-wave celerity ($C_G$), obtained by comparing inlet and outlet hydrograph gravity centres, is constant around 1.3 m.s$^{-1}$, regardless of geology type. This means that, in karst catchments, storm events tend to have longer $T_{Ri}$, $T_{08}$ and $T_{Re}$ values, without slowing down of flood-wave routing, corresponding to an increase in the diffusivity of the flood wave at constant celerity. However, celerity shows a regional pattern, consistent with mean regional reach slopes (Fig. 5), with values increasing

from 0.8 m.s$^{-1}$ in Normandy to 1.6 m.s$^{-1}$ in the Cévennes, Jura catchments showing intermediate values.

Figure 7 shows the peakflow evolution towards catchments with karst. K catchments globally align on the first bisector, showing in most cases low peakflow amplification. NK catchments mostly lie above the equation line of y = 1.5x, showing an increase of at least 50% in specific peakflow, meaning that lateral inflow is higher in those catchments. M catchments have an intermediate hydrologic response.

To summarize, hydrograph-shape descriptors globally show that during the strongest storm events, karst areas tend to decrease peakflow amplification and increase characteristic flood times, without impacting flood-wave celerity. It should be noted that this general pattern is associated to a large variability of hydrological response in K catchments with locally contrasting behaviour.

## 4.3 Lateral-exchange descriptors

The distribution of lateral-exchange descriptors is shown on Fig. 6 (bottom row). The component of peakflow variation due to channel diffusivity ($\Delta_D/V_P$) is quite stable with geology type, with values around -4 E10$^{-7}$ s$^{-1}$. The component of peakflow variation due to channel lateral exchanges ($\Delta_L/V_P$) correlation with geology type is more significant, with median values increasing from 1 E10$^{-6}$ s$^{-1}$ to 2.5 E10$^{-6}$ and 5 E10$^{-6}$ s$^{-1}$ for K, M and NK reaches, respectively. This shows that the strongest lateral inflow in intermediate catchments occurs in NK areas, as expected from the $V_P$ and $V_Q$ variability (Fig. 5), as well as

following $Q_{o,x}$ vs. $Q_{I,x}$ (Fig. 7). It also highlights the compensation of diffusivity peakflow attenuation by lateral inflow, which is stronger in NK catchments. Lateral exchanges are negative for the first quartile of events on K reaches, against very few events for NK reaches. Karst influence on $\Delta_D/V_P$ and $\Delta_L/V_P$ is similar in Jura and Normandy catchments, whereas Cévennes catchments show higher peakflow variations in both components, leading to a strong variability in $\Delta$ values (Fig. 6).

This trend is confirmed by the volume of lateral inflows ($V_{L+}/V_P$) values that decrease with karst occurrence. Volume of lateral

outflows ($V_{L-}/V_P$) distribution also varies with geology type, lateral outflow being reduced in NK catchments. Finally, both descriptors undergo similar karst influence in the three study areas, with only Cévennes K catchments having higher lateral outflow.

Analysis of the simulated lateral hydrographs shows that the weak peakflow amplification of karst reaches is mostly due to a low exchange component $\Delta_L$, associated with a peakflow attenuation caused by diffusivity $\Delta_D$. This is confirmed by lateral

$V_{L+}/V_P$ inflow that is three times lower in K catchments than in NK catchments.

Figure 6 also shows the distribution of IGF*/$V_P$ (IGF* = IGF + $\delta$), which indicates that storm events over NK catchments are mostly characterized by incoming IGFs (>75% of all events). K and M catchments are subject to more balanced incoming and outgoing IGFs, with a median IGF*/$V_P$ value of 0.06 against 0.2 for NK. Absolute IGF* depths for K and M catchments are mostly <50 mm per storm event, against 100 mm or more in NK catchments (results not shown).

IGF*/$V_P$ distribution highlights some differences for K catchments in each study area. Storm events in Jura K catchments mostly show streamflow excess (>75% of all events), with a mean IGF*/$V_P$ value of 0.1. Cévennes and Normandy K catchments have a more balanced behaviour, with median IGF*/$V_P$ values close to zero and storm events resulting in streamflow excess or -deficit in comparable proportions.

### 4.4      Statistical tests

Table 2 shows for each descriptor the results of 't' tests for all combinations of paired samples (Sect. 2.5). The quick- and slow-flow components $V_S$ and $V_U$ both show statistically significant variations with geology type (increasing from K to NK) and with study area (Fig. 6). Despite visible trends in $R_C$, $R_{C,S}$ and $S_C$ variation with geology type (Fig. 6), this correlation has a low statistical significance, with only $R_{C,S}$ being discriminant between K and NK catchments. $R_C$ and $R_{C,S}$ on K catchments show significant variation with study area, which was not verified for $S_C$.

Storm-hydrograph shape is strongly related to geology type, with peakflow amplification reduced and locally attenuated. The longer characteristic times for karst catchments (Fig. 6) have a statistical significance regarding characteristic times. Only flood-wave celerity is not affected by geology type. Karst influence is area-specific for $T_{Ri}$ and $T_{08}$ characteristic times (increasing order of characteristic times is C to N to J).

     Lateral-exchange descriptors show that peakflow amplification in karst reaches is limited, mostly due to its exchange

component $\Delta_L$ being reduced (Fig. 6). This is confirmed by the volumes of lateral exchange, $V_L$/$V_P$ showing that K and M catchments are more prone to lateral outflow than NK catchments, and $V_{L+}$/$V_P$ showing that NK catchments are more prone to lateral inflow than K and M catchments. IGF*/$V_P$ also seems related to geology type. Table 2 shows that the statistical significance of these relationships is not systematic, M catchments being generally different from NK ones. Only the IGF*/$V_P$ descriptor does not show a statistical distinction between all samples.

## 420   5      Discussion

### 5.1      Factors driving flood processes

     The analysis of descriptor distribution (Fig. 6) and its statistical significance (Table 2) shows that several factors affect flood processes at the elementary catchment scale. Water-balance descriptors show that quick- and slow-flow depths $V_S$ and $V_U$ vary with geology type in a statistically significant way. This is partly explained by the role of karst in how catchments transform

precipitation to runoff ($R_C$ and $R_{C,S}$), and on the relative proportions of quick- and slow-flow ($S_C$). Nevertheless, only the $R_{C,S}$ test is statistically significant according to geology type. This highlights the role of climatic influence, as $V_S$ and $V_U$ also depend on rainfall depth $V_P$ that varies with study area, but their range is clearly higher than that of $V_P$.

     In addition, hydrograph descriptors show a reduction in peakflow amplification on K catchments in all study areas, associated with an increase of characteristic times. This trend is consistent with previous work showing peakflow attenuation in karst

areas (De Waele et al., 2010; Charlier et al., 2019). The influence on characteristic times is area-specific, Jura catchments having greater inertia than Cévennes ones. This area-dependent nature of characteristic flood times may be partly explained by different karst settings and occurrence, but is also linked with rainfall patterns. Cévennes catchments have a Mediterranean climate, with typical intense and short storm events triggering flash floods (Marty et al., 2013). The inertial events of Jura catchments can be explained by their great length compared to other areas: Cévennes and Normandy rivers hardly reach 150 km

before reaching the sea or the Rhône, but the downstream Jura station is 400 km from the source along the Doubs river.

     Finally, lateral-exchange descriptors show contrasting exchanges between different geology types, with more lateral inflow in NK catchments and more lateral outflow in K catchments. This agrees with previous studies on storm events in karst catchments highlighting river losses in karst reaches (Delrieu et al., 2005; Perrin and Tournoud, 2009; Bailly-Comte et al.,

2012; Charlier et al., 2019). This trend accompanies a study-area variation, where the main aspect is a higher variability of exchanges for Cévennes catchments, and a lower one for Normandy. This is probably related to the need of harmonizing space scales between gain/loss processes and gauging networks. In fact, earlier work (Toth, 1963; Schaller and Fan, 2009; Bouaziz et al., 2018; Fan, 2019) shows that the size of the investigated catchment affects the IGF importance, greater areas being more likely to be self-contained. In the case of our study areas, as the median gauged areas are similar for each of them (Table 1), it might be explained by the generally thick soil and epikarst in Normandy, which is very reduced in the mountainous and Causses areas of Cévennes. Indeed, soil and epikarst are more likely to promote subsurface flow with closer zones of gains and losses, whereas exposed karst drains, as in the Cévennes, enlarge the space scale of IGF processes by connecting river losses to trans-catchment karst aquifers.

Hydrological processes have been shown to be influenced by physiographic parameters such as karst occurrence. Nevertheless, other drivers can control catchments hydrological response to storm events. As an example, figure 8 shows IGF* depths distribution for the three studied sites, according to seasons. Seasons have been selected in order to reflect the main periods of the hydrological year, and to be suitable for the three sites. Throughout the hydrological year (from September to August), median IGF* depth is continuously decreasing, for all sites towards zero. This is probably linked to the hydrological conditions of catchments, in particular the saturation state of aquifers. Indeed, low water table periods (Apr to Aug) are more likely to limit IGFs. In the case of Jura catchments, a majority of events show IGF gains during high-flow periods (fall and winter), whereas a majority of events promote IGF losses during low-flow periods. This reversal of IGF direction, in the particular case of karst catchments, can also be favoured by the complex organization of the underground conduit networks. In fact, connection or disconnection of the main networks could temporally modify the hydrogeological catchment boundaries with threshold effects (Charlier et al., 2012; Bonacci, 2015). For those reasons, spatial and temporal variability of karst catchments behaviour are still challenging to characterize and predict. A comprehensive understanding of flood processes would thus imply accurate and continuous monitoring of climatic, hydrological and hydrogeological variables, on multiple catchments covering various physiographical settings.

## 5.2    Regional patterns and typology

Figure 6 shows that storm events in K catchments in the Cévennes are mostly associated with losing IGF*s (that can include storage variation), whereas Jura events are mostly associated with gaining IGF*s. This can be linked to the typology of karst settings. The Cévennes region is characterized by binary karst systems, with large upstream NK (hard-rock) terrains and downstream limestone plateaus where karst influence occurs. In such a geological setting, karst areas mostly play a role of flood attenuation as they lie downstream of reliefs with intense rainfall events and high runoff coefficients (see the example of the Tarn river in Charlier et al., 2015). The Jura Mountains region is regionally much more homogeneous, with widespread karst formation affecting the limestone plateaus, and few areas covered with Quaternary deposits. In this setting, karst can alternately promote streamflow capture (attenuation) and generation (amplification), depending on the location of river losses and the interaction between surface-water and groundwater (Le Mesnil et al., 2020).

To discuss the spatial variability of IGFs, the median storm-event IGF* value [mm] is shown for each elementary catchment of the three studied areas on Fig. 9d to 9f, along with the geological maps (Fig 9a to 9c). In the Cévennes (Fig. 9, left row), downstream karstified parts can be divided into two zones defined by a different lithology. The eastern part is mostly underlain by Cretaceous limestone and Cenozoic formations, whereas the western part is mostly composed of (older) Jurassic limestone. Despite IGF* accounting for the potential storage variation (IGF* = IGF + δ), all eastern Cretaceous catchments have negative median IGF* values, meaning that almost certainly groundwater flows out through karst aquifers during storm events, without being recovered in the investigated area during the considered event. The western Jurassic catchments have a less significant

negative IGF*, with most values comprised between -1.45 and 3.5 mm. Through the lithology, this highlights the local role of karst occurrence, which is superimposed on the regional role of general karst settings.

In the Jura Mountains along the main Doubs river (Fig. 9, second row), almost all elementary catchments have negative IGF* during storm events, contrary to the tributary catchments that mostly have positive IGF*. This is coherent with the well-known Doubs river capture, at least for the south-eastern catchments. The tributaries being upstream and higher above the base-level, river loss into karst aquifers is less important or (partly) recovered within a short distance. Storm-event IGF mapping highlights the already well documented zone of Doubs losses feeding the Loue catchment (e.g. Charlier et al., 2014).

Normandy catchments (Fig. 9, third row) show less important IGF* depths, most catchments having values close to zero. This is the result of a too widespread gauging network compared to the spatial scale of IGF processes here. River-capture phenomena are well known in some of the catchments (e.g. Charlier et al., 2019), but the gaining and losing zones mostly seem to fall in the same catchment due to gauging station locations (Sect. 5.1).

Figures 9g to 9i show the percentage of area variation of the elementary catchments, from the topographic catchment area to the fictive area of the hydrogeologically active catchment (i.e. without excess or deficit in water balance). The area of such catchments is calculated from the IGF* and $V_P$ depths. Assuming a spatially homogeneous rainfall intensity, the area variation corresponds to the surface to be added or withdrawn, in regards to the topographic elementary catchment, to obtain the median IGF* value. These variations reflect the incoming or outgoing IGF processes, respectively. For Cévennes and Jura catchments, the values of area variation are of similar magnitudes, ranging from approximately -7 % to +90 %. Regarding Normandy catchments, area variations are slightly lower, with values ranging from -20 % to +32 %. This shows that IGF can be an important term of the water balance during storm events, and has to be considered as such, in order to better understand flow processes during floods.

Figures 9j to 9l show the same elementary catchments area variation, expressed in km². These values are correlated to the size of topographic catchments. They allow a concrete representation of the recharge areas located outside of the surface catchments. As this study is based on storm-events only, and considering that the studied catchments extend further downstream, the sum of fictive active catchments (which is a first approximation of the hydrogeological catchment) do not match with the topographic catchment area. Nevertheless, a previous study by Le Mesnil et al. (2020) showed that, along the Doubs river, annual IGF take a part in the water balance that progressively decreases from spring to outlet and tends towards zero.

## 5.3 IGF: annual vs event scale, impacts on quick- and slow-flow components

In this section, we compare a major missing term of the water budget, alternately estimated by two approaches. First, at the annual scale: the multi-annual IGF (noted $IGF_A$) calculated by Le Mesnil et al. (2020), under the assumption of nil annual stock variation in the aquifer ($\delta = 0$), which is often verified when using several years long data time series. Second, at the event scale: IGF* = IGF + $\delta$ calculated here as defined in Sect. 2.3.3.

Le Mesnil et al. (2020) assessed multi-annual values of $IGF_A$ depth for the catchment of the present work. This $IGF_A$ value, calculated by adapting the two-stage precipitation partitioning theory of Lvovich (1979), is associated with a parameter 'α', estimating the relative impact of IGF on the rapid- and slow-flow components at an annual scale. The α values between 0 and 0.5 correspond to an annual $IGF_A$ mostly affecting the slow-flow component, whereas α values between 0.5 and 1 correspond to an annual $IGF_A$ mostly affecting the rapid-flow component; α values below 0 and over 1 indicate compensating $IGF_A$ flow, such as surface loss combined with groundwater gain. To discuss the link between IGF estimates at both annual and storm event time scales, Fig. 10 presents the storm-event IGF* depths calculated for the present work, as a function of the annual $IGF_A$ depth obtained in the precedent paper for the 108 elementary catchments. Annual $IGF_A$ values were normalized by event duration for consistent comparisons.

Figure 10(a) shows some points with opposed annual IGF$_A$ and event IGF* signs (e.g., a losing annual water-balance-derived IGF$_A$ associated with a gaining event-derived IGF*). The point cloud does not show significant axial organization, leading to a low determination coefficient value for the linear regression. Figures 10(b) and 10(c) present the same relationships for catchments with α values between 0.5 and 1, i.e. annual IGF$_A$ mostly affecting the quick-flow component (Le Mesnil et al., 2020). The R² value of the linear regression has increased but is still low (0.222, not shown). Nevertheless, most catchments show consistent annual storm-event IGF* signs, falling into the upper-right or lower-left quadrants. This means that the annual estimation is good for assessing the relative impact of IGF on quick- and slow-flow components. Moreover, IGF* maps (figure 9) are in accordance with IGF$_A$ results of Le Mesnil et al. (2020). For example, the well-known phenomenon of Doubs river losses feeding Loue catchment is visible at the event scale.

R² values on Fig. 10(b) and 10(c) show that correlations are more reliable when operating regression on groups (by geology type and study area) of K and M catchments. The two groups with a reliable relationship between annual and event-derived IGF depths are the karst catchments K and the Jura catchments J, with respective R² values of 0.664 and 0.663. The stronger relationship in K catchments can be explained by the occurrence of IGF during both recession and flood periods, whereas IGF occurred mostly during flood periods in NK catchments. This agrees with the high positive IGF* values (>50 mm per event) associated with low annual IGF$_A$ values for NK. This may also explain the higher R² value of Jura catchments, as they are mostly K catchments, while the Cévennes also have NK catchments resulting in a lower R² value.

Figure 11 represents V$_S$ versus IGF* (both expressed in depth) for all events and reaches, with different symbols according to geology type. V$_S$ being equal to V$_{S,O}$ – V$_{S,I}$, the V$_S$ versus IGF* relationship provides an insight into the relative proportions of total IGF* depth and quick-flow component variation. First, while IGF* values vary from -200 to 300 mm, V$_S$ values are mostly positive, ranging from -20 to 150 mm. This indicates that for most events, when outgoing IGFs (negative IGF* values) occur, the quick-flow component still increases. This is particularly true on NK catchments, whereas some events with negative IGF* on K catchments show negative (or low positive) V$_S$ values. In that case, a likely hypothesis is the presence of losses in karst riverbeds via sinkholes, which are not present in non-karst reaches (see such example in Charlier et al., 2019). Regarding events with positive IGF*, V$_S$ depths are mainly inferior to IGF* ones, with values around half of the total IGF* ones. This shows that incoming IGFs during storm-events feed the slow-flow component of the total streamflow in similar proportions than the quick-flow component. This is linked to the different IGF processes occurring at the catchment scale: slow-flow component corresponding to aquifer drainage by the river (e.g. Bailly-Comte et al., 2012) and quick-flow component corresponding to karst spring activation (Bonacci and Bojanic, 1991; Maréchal et al., 2008), for example.

**Conclusions**

We carried out a spatialized analysis of 15 easily calculable descriptors characterizing water balance, hydrograph shape and lateral exchanges, for a set of 20 storm-event data at the elementary catchment scale for each of the 108 gauging stations, controlling karst and non karst regions. The results show that karst promotes higher water infiltration, this water being mostly retained during storm events. Karst increases characteristic flood times and limits peakflow amplification, without much affecting flood-wave celerity. This is interpreted to be due to an interbasin groundwater flow (IGF) loss that can be high at the storm-event scale, representing around 50% of the discharge at a catchment outlet, and 20% of rainfall. A spatial variability of those effects is linked to differences in karst regions: binary karst catchments mostly attenuate floods whereas extended karst plateaus undergo alternated losses and gains. Secondary factors include climatic influence (regional variability of rainfall-event intensity), and the spatial-scale match between gain/loss processes and spacing of the gauging network. A seasonal effect has also to be considered regarding IGF magnitude and direction.

Existence of karst hydrological specificities has been known for decades, but is not quantified in a large extent, especially regarding its impacts on flood processes at the scale of the river reach. Though some research had been done on this topic, it

leads to hindered modelling performance in many cases. We have quantified several important parameters for a large set of catchments, for the first time in a spatialized study based on event-scale processes, contributing to build a common understanding at regional scale of karst behaviour during storm events, thus improving modelling and forecasting capabilities in such terrains. Though our approach is based on karst areas it stays generic, and we hope future work will investigate other

relationships between the hydrological response and physiographic characteristics of catchments, such as soil types, land use, climate, etc.

**Author contribution**

RM and JBC were involved in conceptualization, funding acquisition and supervision. MLM gathered the data and designed the methodology with the help of JBC, RM and YC. MLM prepared the manuscript with the help of all co-authors.

**Acknowledgements**

The work was funded by the French Governmental Administration for Risk Prevention (DGPR), the *Service Central d'Hydrométéorologie et d'Appui à la Prévision des Inondations (SCHAPI)*, and the French Geological Survey (BRGM). Hourly streamflow data were gathered from the "Banque hydro" database using a script available at http://doi.org/10.5281/zenodo.3744183.

The authors thank the Editor Jim Freer and two anonymous Reviewers for their constructive comments.

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

**Figures**

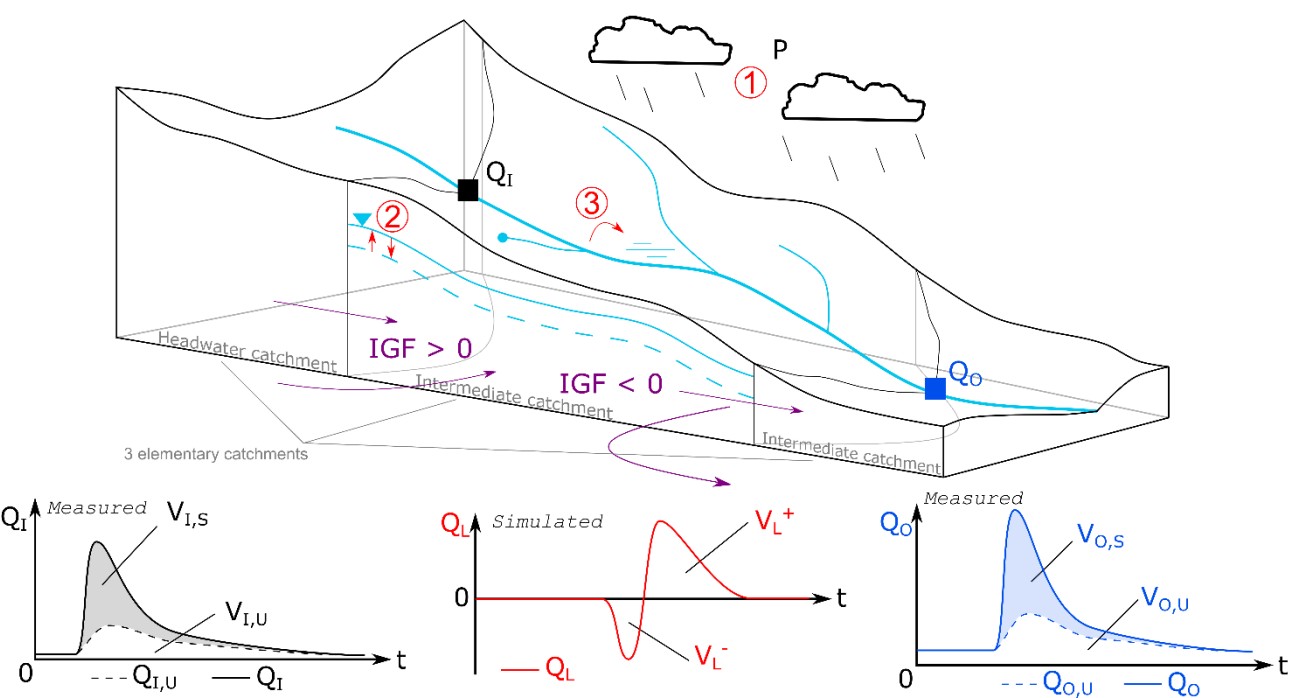

**Figure 1 : Main hydrological flow types at the elementary catchment and storm-event scales, with corresponding measured and simulated hydrographs. Flow definitions are given in Eq. (1) to (5). The corresponding volumes, integrated for the storm event, are noted V with the same indices (see Appendix D for symbols). Lateral flow $Q_L$ is a combination of IGF, effective rainfall (1), aquifer storage variation (2) and overbank phenomena (3, neglected here).**


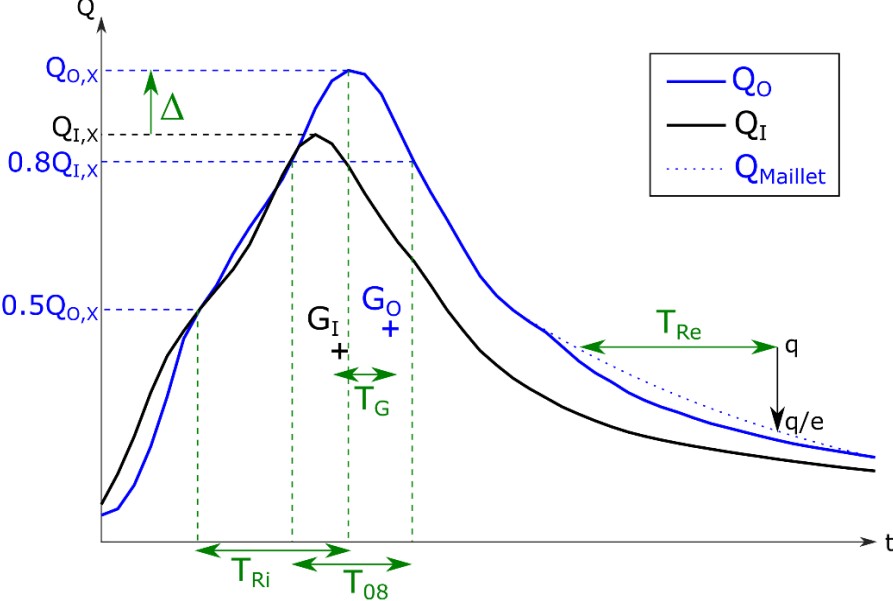

**Figure 2: Storm hydrograph with characteristic times and discharge values. $Q_{O,X}$ and $Q_{I,X}$ are peakflows of outlet and inlet station hydrographs $Q_O$ and $Q_I$, respectively. $T_G$ is the elapsed time between the corresponding gravity centres $G_O$ and $G_I$.**

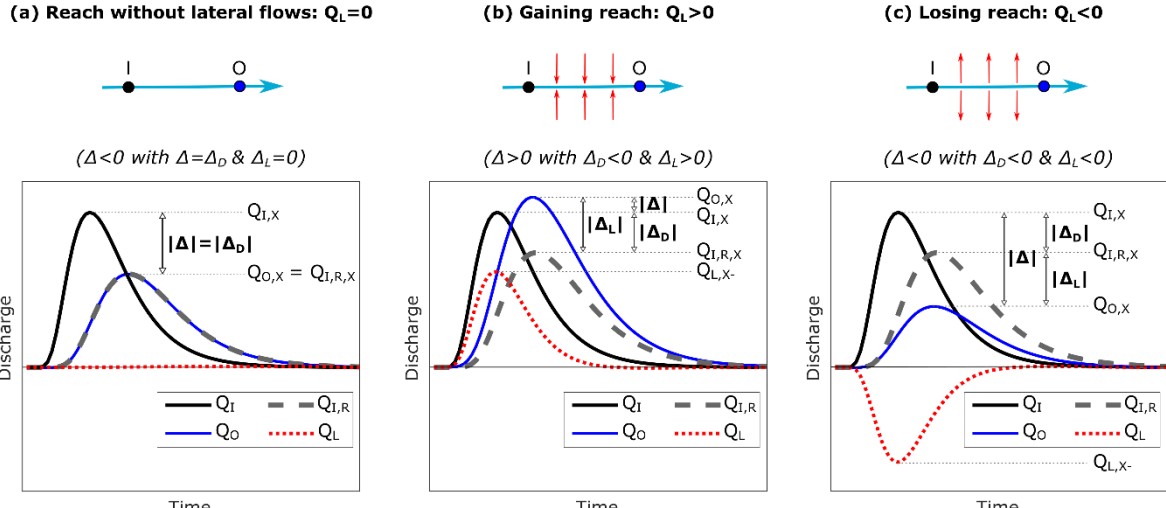

**Figure 3: Theoretical examples of simulated lateral hydrographs $Q_L$, with corresponding values of peakflow variation $\Delta$ and its two components $\Delta_D$ and $\Delta_L$.Modified from Charlier et al., 2019. $Q_{O,x}$ and $Q_{I,x}$ are peakflows of the outlet and inlet station hydrographs $Q_O$ and $Q_I$, respectively. $Q_{I,R}$ is the theoretical outlet flow without lateral exchange (routed inflow by DWE, with C and D parameters).**

**Figure 4: (a): Location of the studied areas with karst regions in France (from BDLISA database). (b), (c), (d): Distribution of gauging stations and geology type of elementary catchments. Red circles indicate gauging stations in intermediate catchments; green circles indicate gauging stations in headwater catchments.**

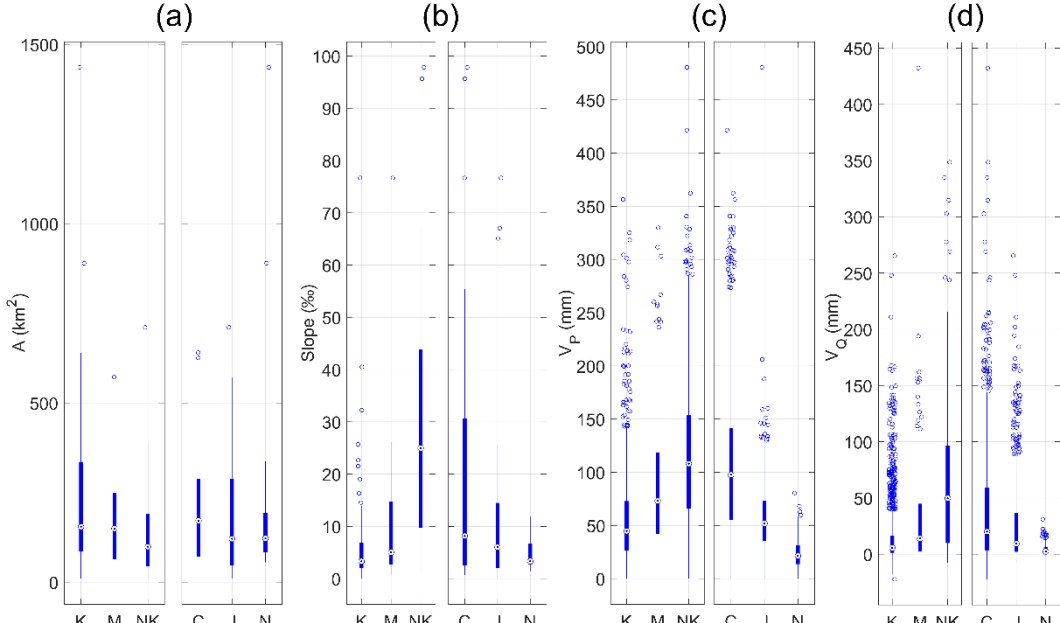

**Figure 5: Elementary catchment area (a); reach mean slope (b) distribution of the 108 gauging stations, grouped by geology type (K: karst, M: mixed, NK: non-karst) and by study area (C: Cévennes, J: Jura, N: Normandy) for K catchments. Precipitation (c) and runoff (d) depth distribution for the 20 selected storm events on the 108 gauging stations, grouped by geology type and study area for K catchments. Values beyond dashed lines are represented on the lines.**

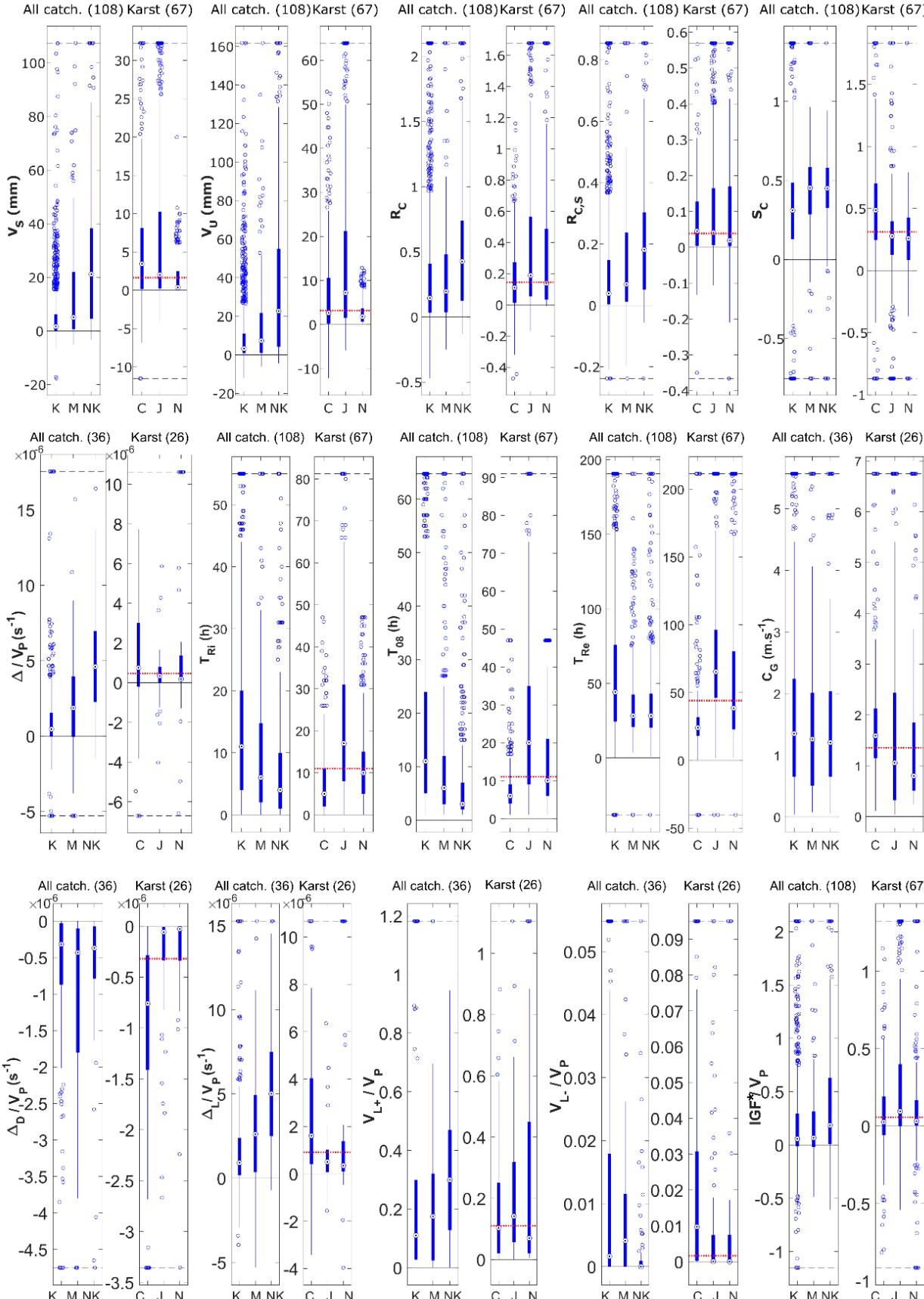

**Figure 6: Distribution of hydrological descriptors, grouped by geology type (K: karst, M: mixed, NK: non-karst) and by location for karst catchments (C: Cévennes, J: Jura, N: Normandy). First row: water-balance descriptors; second row: hydrograph-shape descriptors; third row: lateral-exchange descriptors. Red dotted lines show the median value of karst catchments for the whole sample. Values beyond black dashed lines are represented on the lines.**

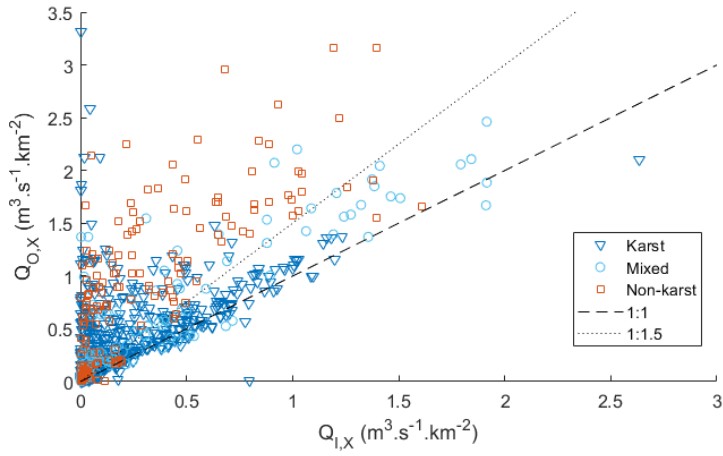

**Figure 7: Variation of peakflow (Qx, see Appendix D) from inlet to outlet stations, with representation of geology type.**

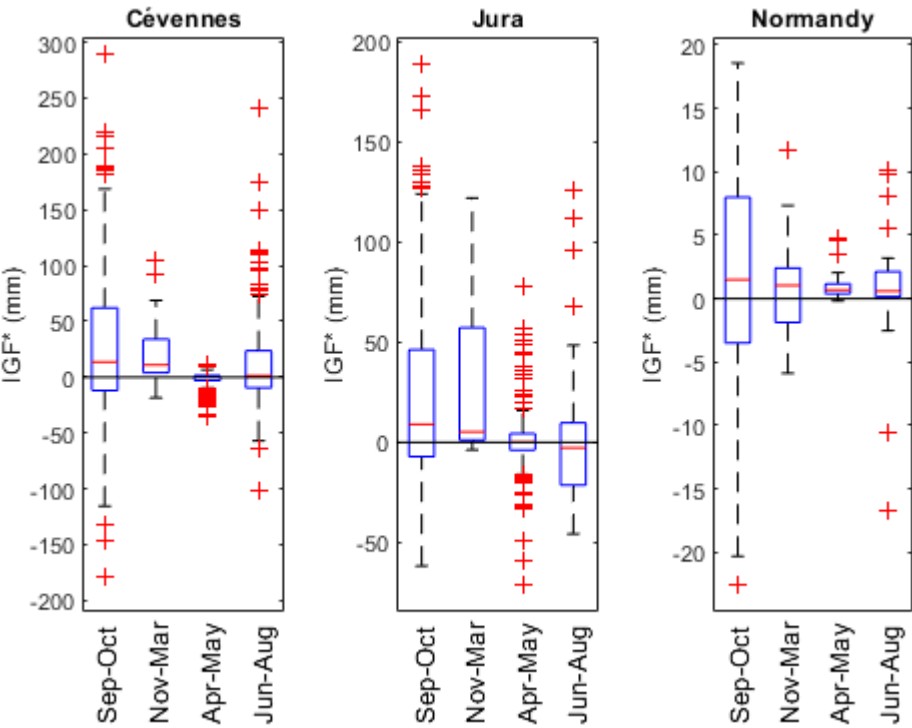

**Figure 8: IGF\* depths distribution for the three studied sites, according to seasons**

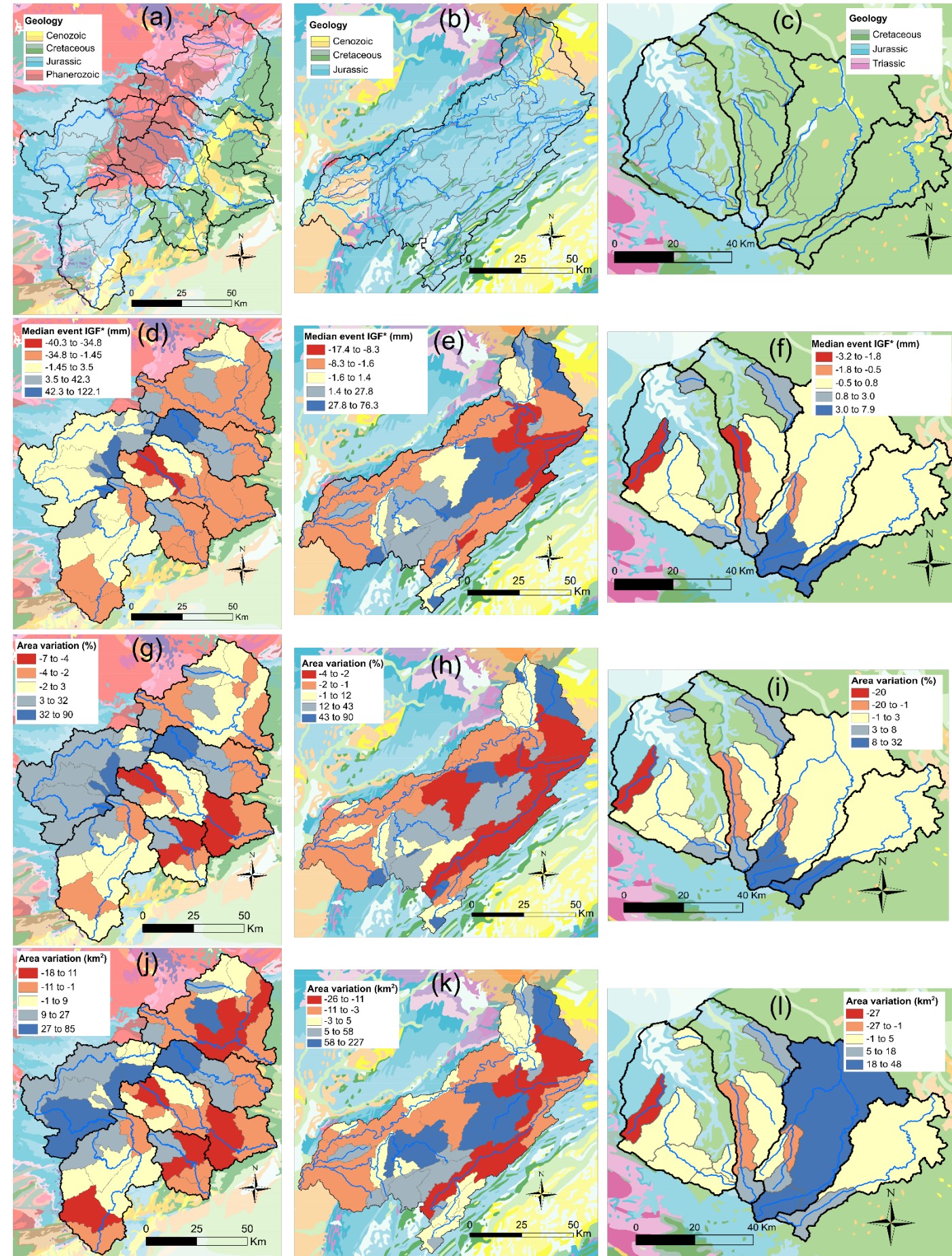

**Figure 9: Main geological features in the three studied areas (a, b, c), median storm event IGF\* values (including potential aquifer storage variation) (d, e, f), and area variation from actual topographic to virtual hydrogeologically active (in %) (g, h, i) and in km² (j, k, l).**

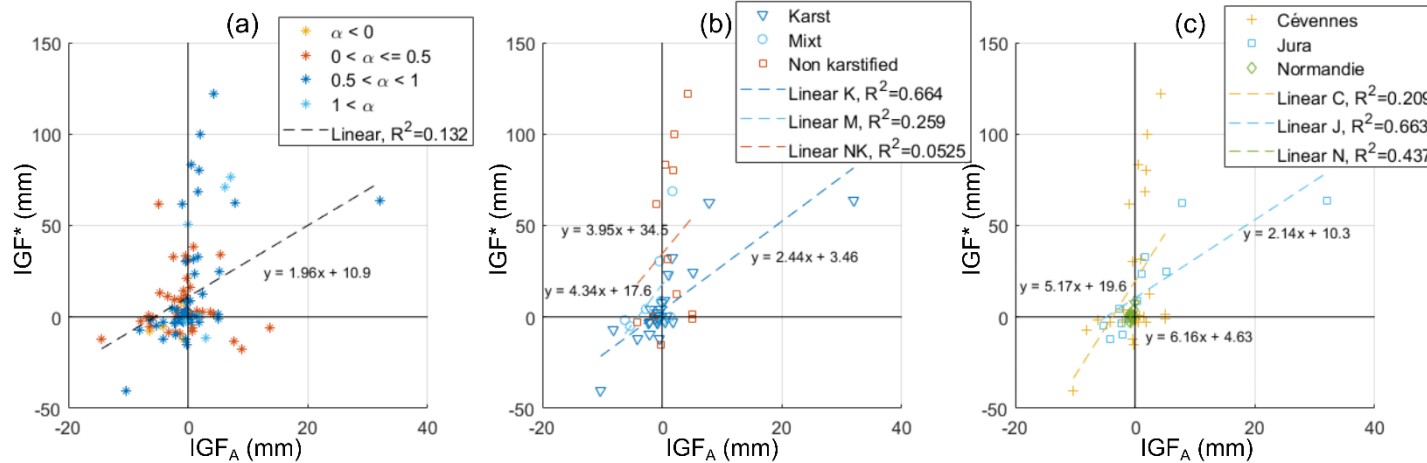

**Figure 10: Relationships between multi-annual water-balance derived IGF_A and storm-event derived IGF\* (including potential aquifer storage variation) depths. IGF_A depth is normalized by event duration for consistent comparisons. First, all 108 elementary catchments were plotted (a), then catchments were filtered to retain only α values between 0.5 and 1; these were plotted grouped by geology type (b) and study area (c).**

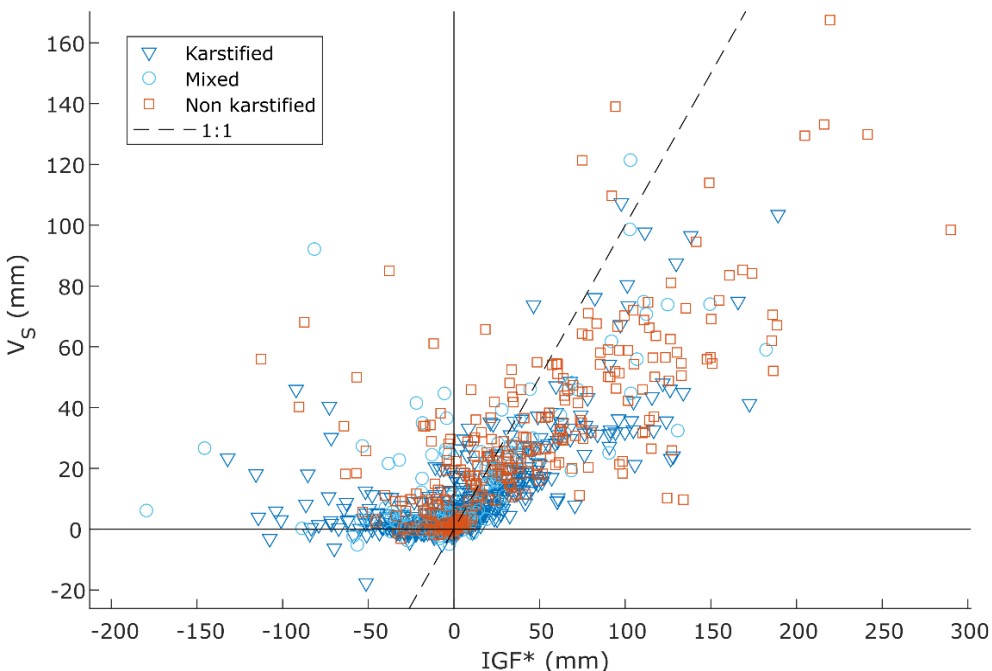

**Figure 11: V_S depths versus IGF\* depths for all events and reaches, with differentiation of geology types**

**Table 1: Studied catchments and associated available data**

|  | Study zone area (km²) | Gauging stations | Median gauged area (km²) | Time series length |
|---|---|---|---|---|
| Ardèche | 2,257 | 9 | 193 | 1996 – 2018 |
| Cèze | 1,048 | 6 | 192 | 2002 – 2018 |
| Gardons | 1,853 | 10 | 137 | 2008 – 2018 |
| Vidourle | 772 | 4 | 182 | 2009 – 2018 |
| Hérault | 2,203 | 9 | 223 | 2007 – 2018 |
| Tarn | 2,145 | 13 | 64 | 1984 – 2018 |
| Total Cévennes | 10,300 | 51 | 172 | 13 years (median) |
| Doubs | 7,400 | 39 | 121 | 1998 – 2018 |

| | | | | | | |
|---|---|---|---|---|---|---|
| Total Jura | 7,400 | 39 | 121 | 21 years | | |
| Iton | 1,048 | 2 | 524 | 1999 – 2018 | | |
| Risle | 1,803 | 5 | 84 | 2001 – 2018 | | |
| Touques | 800 | 5 | 106 | 2010 – 2018 | | |
| Dives | 879 | 6 | 113 | 2009 – 2018 | | |
| Total Normandy | 6,800 | 30 | 127 | 13 years (median) | | |
| Total all basins | 24,500 | 120 | 145 | 15 years (median) | | |

**Table 2: Synthesis of t test decisions for the null hypothesis that data in paired tested samples come from independent random samples from normal distributions with equal means and equal but unknown variances. The result is '1' if the test rejects the null hypothesis at the 5% significance level, and '0' otherwise; p-values are indicated with letters (a = p<0.001, b = 0.001<p<0.01, c = 0.01<p<0.05). The test is performed i) in the three study areas for all combinations of paired samples by geology type (K: karst, M: mixed, NK: non-karst), and ii) for all combinations of karst-catchment paired samples by study area (C: Cévennes, J: Jura, N: Normandy).**

| Descriptor | K-NK | M-NK | M-K | C-J | C-N | J-N |
|---|---|---|---|---|---|---|
| Water-balance descriptors | | | | | | |
| $V_S$ | 1a | 1a | 1a | 1c | 1a | 1a |
| $V_U$ | 1a | 1a | 1a | 1a | 1a | 1a |
| $R_C$ | 0 | 0 | 0 | 1a | 1b | 1b |
| $R_{C,S}$ | 1c | 0 | 0 | 1a | 1b | 1b |
| $S_C$ | 0 | 0 | 0 | 0 | 0 | 0 |
| Hydrograph-shape descriptors | | | | | | |
| $\Delta/V_P$ | 0 | 1a | 0 | 1b | 1c | 0 |
| $T_{Ri}$ | 1a | 1b | 1a | 1a | 1a | 1a |
| $T_{08}$ | 1a | 1a | 1a | 1a | 1a | 1a |
| $T_{Re}$ | 1b | 0 | 1c | 0 | 0 | 0 |
| $C_G$ | 0 | 0 | 0 | 0 | 1c | 0 |
| Lateral-exchange descriptors | | | | | | |
| $\Delta_D/V_P$ | 0 | 1c | 1c | 1a | 1b | 0 |
| $\Delta_L/V_P$ | 0 | 1b | 0 | 1a | 1c | 0 |
| $V_{L+}/V_P$ | 0 | 1c | 0 | 0 | 1b | 1c |
| $V_{L-}/V_P$ | 0 | 1b | 0 | 1b | 0 | 0 |
| $IGF*/V_P$ | 0 | 0 | 0 | 1a | 0 | 0 |

**Appendix A: Hydrograph decomposition**

At each gauging station, discharge values were filtered in order to separate the quick (stormflow, $Q_S$) and slow (underground flow, $Q_U$) flow components. The quick is traditionally interpreted as the surface component, and the slow one as baseflow, which is that part of streamflow corresponding to aquifer drainage. However, in the case of karst catchments, aquifer drainage can also produce a quick-flow signal because of short transfer times through conduits. Several baseflow-separation methods exist, most being based on graphical analysis, like the fixed-interval, sliding-interval, local-minimum, or Wallingford methods (Gustard et al., 1992; Sloto and Crouse, 1996; Rutledge, 1998; Piggott et al., 2005). Numerical approaches have also been developed (e.g. Lyne and Hollick, 1979; Eckhardt, 2005). In this study, we used an automation of the one-parameter recursive digital filter proposed by Lyne and Hollick (1979), implemented in the HydRun package (Tang and Carey, 2017). The filter equation is defined as:

$$Q_S(t) = \beta Q_S(t-1) + \frac{1+\beta}{2}[Q(t) - Q(t-1)] \tag{A1}$$

with $Q_S(t)$ and $Q(t)$ the filtered quick-flow component and total streamflow at time t, respectively, and $\beta$ the filter parameter. We chose this method as it provides consistent results, like those obtained with graphical approaches (results not shown). It can easily be automated and has only one $\beta$ parameter, fixed at 0.91 after a trial-and-error analysis on the studied catchments, and considering the results of Nathan and McMahon (1990) for 186 catchments.

**Appendix B: Lateral flow simulation using the diffusive wave equation**

**Diffusive wave equation**

An inverse modelling approach is adopted for simulating lateral flow between two gauging stations. This approach simulates the lateral flow $Q_L$, based on measurements from two gauging stations $Q_I$ and $Q_O$.

The diffusive wave equation (DWE), accounting for lateral flow, is an approximation of the St-Venant equation that can be written as:

$$\frac{\partial Q}{\partial t} + C(Q)\left[\frac{\partial Q}{\partial x} - q\right] - D(Q)\left[\frac{\partial^2 Q}{\partial x^2} - \frac{\partial q}{\partial x}\right] = 0 \tag{A2}$$

where x [L] is the length along the channel, t [T] is the time, and celerity C(Q) [LT$^{-1}$] and diffusivity D(Q)[L$^2$T$^{-1}$] are functions of the discharge Q [L$^3$T$^{-1}$]. The term q(x,t) [L$^2$T$^{-1}$] represents the lateral flow distribution. The lateral hydrograph $Q_L(t)$ is given by:

$$Q_L(t) = \int_0^l q(x,t)dx \tag{A3}$$

with l [L] the channel length.

Moussa (1996) extended the solution of the DWE under Hayami's hypotheses (semi-infinite channel, C(Q) and D(Q) constant) to the case where lateral flow is uniformly distributed along the channel. Let I(t) and O(t) be the inlet flow minus baseflow and the outlet flow minus baseflow, respectively:

$$O(t) = \varphi(t) + [I(t) - \varphi(t)] * K(t) \tag{A4}$$

with K(t) the Hayami Kernel function defined as:

$$K(t) = \frac{l}{2(\pi D)^{1/2}} \frac{e^{\left[\frac{Cl}{4D}\left(2 - \frac{l}{Ct} - \frac{Ct}{l}\right)\right]}}{t^{3/2}} \tag{A5}$$

and

$$\varphi(t) = \frac{C}{l} \int_0^t [Q_L(\theta) - Q_L(0)]d\theta \tag{A6}$$

**The inverse problem**

Under Hayami's conditions and assuming that lateral flow is uniformly distributed along the channel, Moussa (1996) proposed a solution of the inverse problem; this enables evaluation of the temporal distribution of lateral flow $Q_L(t)$ over the channel reach by knowing I(t) and O(t). Knowing C, D and l, the lateral flow can be calculated using the following procedure:

$$L(t) = O(t) - I(t) * K(t) \tag{A7}$$

$$K^i(t) = K * K * ... * K \quad (i \text{ times}) \tag{A8}$$

$$\varphi(t) = L(t) + L(t)\sum_{i=1}^{\infty} K^i(t) \tag{A9}$$

and finally the lateral flow $Q_{L,C}(t)$

$$Q_{L,C}(t) = Q_L(0) + \frac{l}{C}\frac{d\varphi}{dt} \tag{A10}$$

## Appendix C: Estimation of effective rainfall

Effective rainfall $P_{eff}$ is estimated using three different approaches, in order to provide a range for this component characterized by major uncertainties. The three approaches are based on the water-budget methods proposed by Thornthwaite (1948) and Dingman (2002), and on the GR lumped model (Edijatno et al., 1999). All three consider soil as a reservoir, used for separating the input (precipitation) into evapotranspiration and effective rainfall. The capacity of the soil reservoir $C_{max}$ is estimated with the map of available soil-water capacity from INRAE (Le Bas, 2018).

In the **Thornthwaite method**, water in the soil reservoir is directly available for evapotranspiration, and precipitation produces effective rainfall ($P_{eff}$) only after soil saturation. The following algorithm summarizes the method:

- If P <E0, the difference E0 – P is subtracted from the soil-water stock B until it is empty:
  - $B_t = max\,(0\,;\,B_{t-1} + P_t - E0_t)$
  - $E_t = min\,(E0_t\,;\,B_{t-1} + P_t)$
  - $P_{efft} = 0$
- If P >E0, the difference P – E0 first feeds the soil-water stock B and then produces efficient rainfall:
  - $B_t = min\,(B_{max}\,;\,B_{t-1} + P_t - E0_t)$
  - $E_t = E0_t$
  - $P_{efft} = max\,(0\,;\,B_t + P_t - E0_t - B_{max})$

The **Dingman method** is similar to the previous one, with an exponential law governing water extraction for evapotranspiration from the soil reservoir:

- If P <E0, the difference E0 – P is subtracted from the soil water stock B following an exponential law:
  - $B_t = B_{t-1} \cdot e^{\frac{-(E0_t - P_t)}{B_{max}}}$
  - $E_t = P_t + B_{t-1} - B_t$
  - $P_{efft} = 0$
- If P >E0, the difference P – E0 first feeds the soil-water stock B and then produces efficient rainfall (as in the Thornthwaite method):
  - $B_t = min\,(B_{max}\,;\,B_{t-1} + P_t - E0_t)$
  - $E_t = E0_t$
  - $P_{efft} = max\,(0\,;\,B_t + P_t - E0_t - B_{max})$

The **GR method** is derived from the GR hydrological models (Edijatno et al., 1999) and involves a quadratic law for the water-level variation in the soil reservoir. The algorithm, summarized below, then was adapted to the BRGM 'Gardenia' model (Thiéry, 2014), which has been used here.

- If P <E0, the difference $E_n$ = E0 – P is subtracted from the soil-water stock B, following a quadratic law:
  - $dB = \left((B/B_{max})^2 - 2(B/B_{max})\right) \cdot dE_n$
  - $dE_t = -dB$
  - $P_{eff} = 0$
- If P >E0, the difference $P_n$ = P – E0 is partitioned into effective rainfall and soil storage following a quadratic law:
  - $dB = (1 - (B/B_{max})^2) \cdot dP_n$
  - $E = E0$
  - $dP_{eff} = (B/B_{max})^2 \cdot dP_n$
- Integration of the differential variations provides expressions of $B_t$, $E_t$ and $P_{efft}$ as a function of $B_{t-1}$, $B_{max}$, and $tanh(E_n/B_{max})$ or $tanh(P_n/B_{max})$.

The final $P_{eff}$ value corresponds to the mean of the three estimation method results.

## Appendix D: List of symbols

| Symbol | Unit | Formula | Description |
|---|---|---|---|
| **Water-balance descriptors** | | | |
| $V_S$ | mm | $(V_{O,S} - V_{I,S}) / A$ | Stormflow depth |
| $V_U$ | mm | $(V_{O,U} - V_{I,U}) / A$ | Baseflow depth |
| $R_C$ | - | $(V_O - V_I) / V_P$ | Runoff coefficient |
| $R_{C,S}$ | - | $V_S / V_P$ | Storm-runoff coefficient |
| $S_C$ | - | $V_S / V_Q$ | Stormflow coefficient |
| **Hydrograph-shape descriptors** | | | |
| $\Delta / V_P$ | $s^{-1}$ | $(Q_{O,X} - Q_{I,X})/V_P$ | Peakflow variation normalized by rainfall |
| $T_{Ri}$ | h | - | Time duration of rise from $0.5 Q_{O,X}$ to $Q_{O,X}$ |
| $T_{08}$ | h | - | Time duration at $Q_O > 0.8 Q_{O,X}$ |
| $T_{Re}$ | h | Eq. (7) | Time constant of recession (exponential model) |
| $C_G$ | $m.s^{-1}$ | $l/(t(G_O) - t(G_I))$ | Celerity (based on hydrograph gravity centres) |
| **Lateral-exchange descriptors** | | | |
| $\Delta_D / V_P$ | $s^{-1}$ | $(Q_{I,R,X} - Q_{I,X})/ V_P$ | Diffusivity peakflow variation normalized by rainfall |
| $\Delta_L / V_P$ | $s^{-1}$ | $(Q_{O,X} - Q_{I,R,X})/ V_P$ | Exchange peakflow variation normalized by rainfall |
| $V_{L+}/ V_P$ | - | $V_{L+}/ V_P$ | Lateral inflow normalized by rainfall |
| $V_{L-}/ V_P$ | - | $V_{L-}/ V_P$ | Lateral outflow normalized by rainfall |
| $IGF^*/V_P$ | mm | $(V_Q - V_{Peff})/ V_P$ | Interbasin Groundwater Flow normalized by rainfall |
| **Other variables** | | | |
| $A$ | m² | - | Catchment area |
| $\beta$ | $h^{-1}$ | - | Recession coefficient as per Maillet approximation |
| $C$ | $m.s^{-1}$ | - | Celerity, parameter of the DWE here taken as $C_G$ |
| $D$ | | - | Diffusivity, parameter of the DWE |
| $\delta$ | mm | - | Aquifer-storage variation, expressed as catchment depth |
| $E$ | mm | Appendix C | Actual evapotranspiration |
| $G_I; G_O$ | - | - | Inlet and outlet hydrograph gravity centres |
| $IGF^*$ | mm | $V_O - V_I - V_{Peff}$ | IGF and aquifer variation = streamflow excess or deficit |
| $IGF_A$ | mm | - | Multi-annual IGF calculated in Le Mesnil et al., 2020 |
| $l$ | m | - | River reach length |
| $P; V_P$ | $mm.s^{-1}$; mm | - | Precipitation rate and depth on the elementary catchment |
| $P_{eff}; V_{Peff}$ | $mm.s^{-1}$; mm | Appendix C | Effective rainfall rate and depth |
| $q$ | $m^3.s^{-1}$ | - | Streamflow at the beginning of recession limb |
| $Q_O; V_O$ | $m^3.s^{-1}$; mm | -; - | Streamflow and water depth at catchment outlet |
| $Q_I; V_I$ | $m^3.s^{-1}$; mm | -; - | Streamflow and water depth at catchment inlet |
| $Q; V_Q$ | $m^3.s^{-1}$; mm | $Q_O - Q_I; V_O - V_I$ | Streamflow and water depth at elementary catchment |
| $Q_{O,U}; V_{O,U}$ | $m^3.s^{-1}$; mm | -; - | Slow streamflow and water depth at catchment outlet |
| $Q_{O,S}; V_{O,S}$ | $m^3.s^{-1}$; mm | -; - | Quick-flow component and water depth at catchment outlet |
| $Q_{I,U}; V_{I,U}$ | $m^3.s^{-1}$; mm | -; - | Slow-flow component and water depth at catchment inlet |
| $Q_{I,S}; V_{I,S}$ | $m^3.s^{-1}$; mm | -; - | Quick-flow component and water depth at catchment inlet |
| $Q_U$ | $m^3.s^{-1}$ | $Q_{O,U} - Q_{I,U}$ | Slow-flow component at elementary catchment |

| | | | |
|---|---|---|---|
| $Q_S$ | $m^3.s^{-1}$ | $Q_{O,S}-Q_{I,S}$ | Quick-flow component at elementary catchment |
| $Q_{O,X}$ | $m^3.s^{-1}.km^{-2}$ | - | Peakflow at catchment outlet normalized by A |
| $Q_{I,X}$ | $m^3.s^{-1}.km^{-2}$ | - | Peakflow at catchment inlet normalized by A |
| $Q_L$ | $m^3.s^{-1}$ | Appendix B | Simulated lateral-exchange flow |
| $Q_{I,R}$ | $m^3.s^{-1}$ | Appendix B | Routed inlet streamflow |
| $Q_{I,R,X}$ | $m^3.s^{-1}$ | Appendix B | Routed inlet streamflow peakflow |
| $Q_{Maillet}$ | $m^3.s^{-1}$ | $q \cdot \exp(-\beta t)$ | Maillet approximation of recession streamflow |
| $T_G$ | h | - | Elapsed time between $G_I$ and $G_O$ |

875