# Peer review of "Impact of karst areas on runoff generation, lateral flow and interbasin groundwater flow at the storm-event timescale"

_Hydrology and Earth System Sciences, 2020_

## Referee Comment (RC1) · Anonymous Referee #1 · 19 Jul 2020

The study analyzed stream flow observations at 108 gages in three regions with varying karst area, focusing on event-scale hydrologic response to 20 large storm events. Using 15 descriptors on catchment water balance, hydrograph shape, and difference between an upstream and downstream gage, the study compared the catchment response in karst (K), middle (M) and non-karst (NK) catchments. It is concluded that (1) karst promotes high infiltration but (2) slows down flood response (both rising and falling limb), and the latter behavior is attributed to inter-basin groundwater flow (IGF).

I find the study valuable in that it selected a wide range of catchments under different climatic and geologic conditions, and that it focused on event time-scale response

of catchment runoff underlain by karst geology. However, the manuscript can benefit significantly from a clearer and sounder conceptual model, better framing of the questions (what we know and don't know) and the hypotheses to be tested with this particular dataset, how the results speak for or against the hypotheses posed, and a better overall presentation, as expanded below.

First, there are some conceptually ambiguities.

(a) My understanding is that groundwater flow, particularly trans-catchment groundwater flow, or inter-basin flow (IBF), does not follow surface drainage structure in karst terrain as depicted in Figure 1. We cannot delineate groundwater basins based on surface topography, a well-known problem in karst terrain. Figure 1 depicts the IBF as completely defined by, and in parallel with the surface drainage gradient. While this may be fine for unconsolidated materials without strong geologic structure, it is hardly the case for karst terrain, where the extensive underground conduit networks do not follow the topography. The authors' finding that the strongest lateral inflow is in intermediate catchments in NK, that is, there is more IBF in non-karst areas, is counter intuitive and confusing, and points to this conceptual flaw. The reason may well be that in NK areas the surface and subsurface drainage system are more aligned, and such an assumption is more valid, allowing the detection of IBG inflow. The authors need to clarify and justify this assumption, because it also underlies the methodology used in the study, with the analyses entirely based on streamflow.

(b) The methodology infers IBF into and out of a catchment by comparing the inflow QI and outflow Qo of a stream reach (Fig 1). The authors need to clarify to what extend one can infer IGFs from streamflow alone. The difference reflects inflow from local precipitation falling on its topographic catchment and subsequent infiltration and surface and subsurface runoff (local source), the inflow from upstream catchments via IBF (remote source), minus the loss to local aquifers (local sink, recharging local aquifers, increasing groundwater storage) which may or may not leave the catchment via IBF (remote sink). But without an explicit aquifer water balance to track all the terms during

the events, it is difficult to separate these terms. To arrive at quantitative results, the authors need to explicitly quantify these terms, incorporating aquifer water level observations, spring discharges. The diffusive wave model for surface water propagation appears to be an over-kill in comparison to a complete lack of first-order water balance in the subsurface.

Second, the authors should provide a more thorough literature review, discuss what we know and do not know in terms of event-scale catchment runoff generation in karst terrain, and pose a set of testable hypotheses. For example, it is expected that in karst area, infiltration is high, and storm flow is flashy due to the open conduits in the subsurface, but after a threshold when the groundwater builds up to the spring overflow point. Then as the results are opposite (slow hydrograph response), one can reason why this hypothesis must be rejected. For another example, one can hypothesize that in karst catchments, streamflow can be lower or higher than expected from calculations based on infiltration over the surface catchment area alone (local source), or there is mass imbalance, and thus IGF must be invoked. Then the analyses can be targeted to test these hypotheses, and the results discussed with clarity surrounding these hypotheses.

Third, in addition to the above, a few other things can be done to reduce the length, enhance focus, clarify terminology and make it easier for readers to follow the central theme and take-home messages.

(a) Move "Section 3 The Study Area" to before "Section 2 Methodology"

(b) List all variables in a table with definitions. It is hard to remember the 10s of mathematical symbols mentioned later – one has to go back and find their definitions.

(c) In presenting results, please use plain English of the meaning of the variable, rather than the ratio of 2 variables defined earlier, so the reader can grasp the meaning of the results.

(d) Is the lateral hydrograph, QL, the same as IGF? Both appeared in the text frequently, and it appeared that they are used in the same context. If so, please unify the terminology. If not, please make it more clear by using a conceptual model diagram.

(e) The authors talk about losing reach vs gaining reach, in the context of losing water to other catchments via IGF. The terms "losing streams" or "gaining stream" have been understood as stream and local groundwater exchange, regardless of the source/sink is local or remote (IGF). Perhaps use "losing catchment" vs "gaining catchment" because here the authors refer to IGF?

(f) in Fig 1, catchments include headwater, intermediate, but is it intermediate between a headwater and a tailwater catchment? Is there another catchment below the intermediate?

Line 296. Is this stream loss, or simply that the infiltration over the surface catchment drained elsewhere, to neighboring catchments? IBF is not just a result of stream flow loss; infiltrated water may not go to the streams at all in its own catchment but may enter streams in other catchments.

---

## Referee Comment (RC2) · Anonymous Referee #2 · 3 Aug 2020

The authors present an interesting study about karst characteristics of catchments at storm-event timescale. The merit of the study is the treatment of multiple sites and the storm-event time-scale, where the case study region in France provides a good example and good data.

The overall article is well written and of scientific interest, but the scientific argumentation should be improved. The literature study is very short and lacking some other approaches, the research question is not clearly stated, the concept is not sufficiently brought into the context of other methods, and the benefits and shortcomings of the developed method against other methods should be better elaborated in the theoretical

part, and later on discussed based on the results.

In the literature study, other concepts for describing the hydrological characteristics of larger scale karst systems should be mentioned. Gárfias-Solizet al. (2010) found system memory, response time and mean delay between input and output as indicators for karstification but emphasized that the physically founded regionalization needs to account for structural complexity, heterogeneity of the lithology and the degree of karstification. Hartmann et al. (2013) described the responsive behavior of large-scale karst systems by system signatures derived from hydrodynamic and hydrochemical observations based on data from five different karst systems. Basha (2020) presented six recession curves for the classification of karst aquifers, describing the dichotomy of the dual flow characteristics between the fissured matrix and the conduit system. How do the descriptors found in this study relate to the indicators and signatures found by other authors? Is there a clear difference between daily and storm-event resolution at the size of catchment under consideration?

"Despite some studies describing significant IGF in karst areas (Le Moine et al., 2007; Lebecherel et al., 2013), the specific issue of IGF in karst has not been addressed as such." There have been studies dealing with IGF, in particular the modelling of IGF at catchment scale or adding IGF to hydrological catchment models. Even if karst-based IGF is often neglected, it found attention among hydrological modelers, introducing or enhancing karst capabilities of e.g. SWAT (e.g., Nguyen et al., 2020). Apart from modelling and data analyses, geo-chemical methods based on analysis of chlorides and stable isotopes has been used to characterize karst systems.

The authors correctly mention that many hydrological studies neglect the characteristics of karst regions, but their own catchment delineation follows the "classical" orographic approach using surface water divides instead of trying to derive subsurface catchments based on hydrogeological data/maps. This is a major shortcoming of the method. In particular, the water balance descriptors could be more meaningful if the overall contributing area including the subsurface catchment would be regarded.

[Figure]

Whenever catchment sizes and rainfall amounts are related with stream flow, there would be a mismatch in karst regions and it is unclear for me how the authors have regarded that aspect. A detailed discussion is missing. In particular, for the description of storm peak flow, I would expect water balance descriptors, which include the areas connected by IGF. For arid basins, Wolaver et al. (2008) developed a method to delineate karstic aquifers based on data. Applying such delineation beyond the usage of surface catchments might help to improve the results of the characterization of the flow response to storm events. Here, one could think that the method is inconsistent and I ask the authors to explain why they did not try to delineate the karstic aquifers and include them into the water balance considerations.

The conclusions seem to ignore the efforts done in karst research over the last decades: "Existence of karst hydrological specificities has been known for decades, but is poorly quantified and documented.". Here, I disagree. The authors should explicitly say, where their method provided methodological innovation and which characteristics of karst regions could be now better described. Literature cited:

Basha, H.A.: Flow Recession Equations for Karst Systems. Water Resources Research 56(7), 2020, Article number e2020WR027384

Gárfias-Soliz, J., Llanos-Acebo, H., Martel, R.: Time series and stochastic analyses to study the hydrodynamic characteristics of karstic aquifers. Hydrological Processes 24(3), January 2010, Pages 300-316.

Hartmann, A. et al.: Process-based karst modelling to relate hydrodynamic and hydrochemical characteristics to system properties. Hydrology and Earth System Sciences 17(8), 2013, Pages 3505-3521.

Nguyen, V.T., Dietrich, J., Uniyal, B.: Modeling interbasin groundwater flow in karst areas: Model development, application, and calibration strategy. Environmental Modelling and Software 124, 2020, Article number 104606.

Wolaver, B.D. et al.: Delineation of regional arid karstic aquifers: An integrative data approach. Ground Water 46(3), 2008, Pages 396-413.

---

## Author Comment (AC1) · 9 Sep 2020

**Response to Anonymous Referee #1**
**(Comment received and published 19 July 2020)**

We are very grateful for the constructive comments of anonymous referee #1 on the manuscript. We believe that they will help increasing substantially the scientific quality of the manuscript. We agree with most recommendations and we give in this note our responses to all remarks.

**Comment 1a)**

The study analyzed stream flow observations at 108 gages in three regions with varying karst area, focusing on event-scale hydrologic response to 20 large storm events. Using 15 descriptors on catchment water balance, hydrograph shape, and difference between an upstream and downstream gage, the study compared the catchment response in karst (K), middle (M) and non-karst (NK) catchments. It is concluded that (1) karst promotes high infiltration but (2) slows down flood response (both rising and falling limb), and the latter behavior is attributed to inter-basin groundwater flow (IGF).

I find the study valuable in that it selected a wide range of catchments under different climatic and geologic conditions, and that it focused on event time-scale response of catchment runoff underlain by karst geology. However, the manuscript can benefit significantly from a clearer and sounder conceptual model, better framing of the questions (what we know and don't know) and the hypotheses to be tested with this particular dataset, how the results speak for or against the hypotheses posed, and a better overall presentation, as expanded below.

Thank you. We propose to modify the manuscript according to your remarks as detailed below. Your review will allow several improvements such as a better justification of our choice to work with topographic catchments as a spatial reference, a more complete literature review, and a clearer definition of what is the lateral flow $Q_L$ in comparison to IGF.

**Comment 1b)**

First, there are some conceptually ambiguities.

(a) My understanding is that groundwater flow, particularly trans-catchment groundwater flow, or inter-basin flow (IBF), does not follow surface drainage structure in karst terrain as depicted in Figure 1. We cannot delineate groundwater basins based on surface topography, a well-known problem in karst terrain. Figure 1 depicts the IBF as completely defined by, and in parallel with the surface drainage gradient. While this may be fine for unconsolidated materials without strong geologic structure, it is hardly the case for karst terrain, where the extensive underground conduit networks do not follow the topography. The authors' finding that the strongest lateral inflow is in inter-mediate catchments in NK, that is, there is more IBF in non-karst areas, is counterintuitive and confusing, and points to this conceptual flaw. The reason may well be that in NK areas the surface and subsurface drainage system are more aligned, and such an assumption is more valid, allowing the detection of IBG inflow. The authors need to clarify and justify this assumption, because it also underlies the methodology used in the study, with the analyses entirely based on streamflow.

First, we propose to keep the term IGF in the manuscript (instead of IBF) as it is a term dedicated to the study of Interbasin Groundwater Flow.

Figure 1 represents all identified inputs and outputs at the scale of an elementary catchment (drained area between two gauging stations). They are composed of three main flows: input streamflow $Q_I$, output streamflow $Q_O$ and lateral flow $Q_L$. $Q_L$ is composed of several flows corresponding to different processes, which are numbered from 1 to 4 on Figure 1. Interbasin Groundwater Flow (IGF) is noted as number 2, and represented by incoming and outgoing arrows of two types. Straight arrows represent IGF in parallel with the surface drainage gradient, while curved arrows represent IGF from/towards other directions (outside upstream or downstream elementary catchments).

→ *Suggested change in manuscript (Figure 1):*  Modify IGF arrows to make clearer the fact that IGF can be in different directions than the main surface drainage direction, and to differentiate them from other $Q_L$ flows.

We fully agree on the comment stated that groundwater basins cannot be delineated based on surface topography, especially in karst areas. However, our spatial reference is the stream catchment for which the topographic catchment is the relevant spatial reference, even when accounting for groundwater flows in addition to surface flows. We propose to modify Figure 1 and to add the following paragraph to our introduction for a better understanding of our strategy.

→ *Suggested change in manuscript (Introduction section): Regional spatial analyses need to be based on reliable data at the highest resolution available. For this purpose, the scale of the elementary catchment - i.e. subdivision of a basin following available gauging stations – appears to be the best resolution for long-term monitoring. Elementary catchment can be either the drained area of a headwater catchment controlled by a gauging station, or the drained area between two gauging stations (intermediate catchment). When considering surface and groundwater components, the delineation method of elementary catchments is questionable (topographic vs. hydrogeological boundaries). Despite the importance of groundwater processes in karst areas, topographic catchment delineation remains a more robust reference, for several methodological reasons. First, IGF can be defined as groundwater flow crossing topographic divides, as this concept emerged with the evidence of certain groundwater systems extending beyond the limits of valleys (Eakin, 1966). A perfectly delineated groundwater basin would then show IGF equal to zero. For this reason, studies related to IGF use the topographic catchment spatial reference (Genereux et al., 2005; Schaller and Fan, 2009; Bouaziz et al., 2018; Nguyen et al., 2020; see also a synthesis in Fan, 2019). Second, although in karst catchments groundwater contributes to flood flow, surface runoff has to be considered as an important component of flood flow. Consideration of hydrogeological catchments could thus lead to wrong surface contribution assessment depending on their surface drainage network. Third, as some groundwater flows are aligned with the main surface drainage axis, hydrogeological catchments would encompass the whole river, making it impossible to study the spatial variability of parameters along the river, at the elementary catchment scale. Finally, topographic delineation is reliable and easily reproducible, while groundwater delineation is characterized by a strong uncertainty and variability in karst areas.*

→ *Suggested change in manuscript (Figure 1):* Add schematic delineations of topographical and groundwater basins.

(b) The methodology infers IBF into and out of a catchment by comparing the inflow QI and outflow Qo of a stream reach (Fig 1). The authors need to clarify to what extend one can infer IGFs from streamflow alone.

Considering the topographic elementary catchment as our reference for all water balance calculations, IGF is inferred from inflow $Q_I$, outflow $Q_o$ and effective rainfall $P_{eff}$ (obtained by subtracting the estimated evapotranspiration to measured rainfall, see Appendix C of the submitted manuscript). Knowing that this term also accounts for potential aquifer level variations, it is noted IGF*.

The difference reflects inflow from local precipitation falling on its topographic catchment and subsequent infiltration and surface and subsurface runoff (local source), the inflow from upstream catchments via IBF (remote source), minus the loss to local aquifers (local sink, recharging local aquifers, increasing groundwater storage) which may or may not leave the catchment via IBF (remote sink). But without an explicit aquifer water balance to track all the terms during the events, it is difficult to separate these terms. To arrive at quantitative results, the authors need to explicitly quantify these terms, incorporating aquifer water level observations, spring discharges.

As explained in lines 93 to 97 and visible in Figure 1, lateral exchanges may be a combination of (supposed decreasing importance):

- Effective rainfall (precipitation minus evapotranspiration) over the elementary catchment,
- IGF,
- Aquifer storage variation,
- Overbank phenomena.

We consider overbank flow to be negligible, assuming that the overflow water returns to the river after a relatively short time during the recession. We propose an assessment of evapotranspiration (explained in Appendix C) which allows estimating the effective rainfall on the elementary catchments. Thus, combined to inlet streamflow $Q_I$ and outlet streamflow $Q_O$ data, our methodology allows calculating the remaining term of the water balance, noted IGF*, which includes IGF and the potential aquifer storage variation during the considered event (noted δ). δ can either be positive, corresponding to an aquifer recharge, or negative, corresponding to an aquifer draining. Aquifer draining is unlikely during storm events, as important rainfalls generally occur. In case of an aquifer recharge, and as our analysis is performed on the whole storm-event period (including the entire recession), a substantial part of the infiltrated water should be released. For these reasons, δ is assumed to be a minor component on the IGF* term compared to IGF. This is discussed in lines 182 to 195, and calculation of IGF* is presented in equation 8.

→ *Suggested change in manuscript (Methodology section):* the text will be made clearer to show how all water balance component are considered in our study.

The diffusive wave model for surface water propagation appears to be an over-kill in comparison to a complete lack of first-order water balance in the subsurface.

We precise that the diffusive wave model is used to analyse the flow dynamics during floods through the simulation of lateral hydrographs. It is not used in the lateral flow estimation for water balance calculations (see our response above).

**Comment 1d)**

Second, the authors should provide a more thorough literature review, discuss what we know and do not know in terms of event-scale catchment runoff generation in karst terrain, and pose a set of testable hypotheses. For example, it is expected that in karst area, infiltration is high, and storm flow is flashy due to the open conduits in the subsurface, but after a threshold when the groundwater builds up to the spring overflow point. Then as the results are opposite (slow hydrograph response), one can reason why this hypothesis must be rejected. For another example, one can hypothesize that in karst catchments, streamflow can be lower or higher than expected from calculations based on infiltration over the surface catchment area alone (local source), or there is mass imbalance, and thus IGF must be invoked. Then the analyses can be targeted to test these hypotheses, and the results discussed with clarity surrounding these hypotheses.

Thank you for this comment. We will develop the literature review (see also our response to referee #2 regarding this aspect), and especially discuss in a larger extent what are the main findings on karst impacts on flood processes and runoff generation. We propose to include the following paragraph to the introduction. This more detailed analysis will also be used to better interpret and discuss our results.

→ *Suggested change in manuscript (Introduction section): Karst impacts on flood processes are mostly documented through case studies. As an example, Zanon et al. (2010) showed that during a flash flood in 2007 in Slovenia, karst area reduced flooding, which was more important in a non-karst neighbouring zone, receiving less precipitation. Likewise, Delrieu et al. (2005) observed, for an exceptional storm event in 2002, lower runoff coefficient values for the karstic catchment compared to the hard-rock catchment in the eastern zone of the Cévennes Mountains. De Waele et al. (2010) and Charlier et al. (2015, 2019) determined that, depending on the location on the river profile, karstification could result in streamflow losses or gains due to the high spatial variability of the hydrogeological karst features. Other frequently described processes are groundwater rising leading to reduced infiltration and important surface runoff (Lopez-Chicano et al., 2002; Bonacci et al., 2006), and backflooding/sinkhole flooding due to conduit constriction (Maréchal et al., 2008; Bailly-Comte et al., 2009).*

**Comment 1e)**

Third, in addition to the above, a few other things can be done to reduce the length, enhance focus, clarify terminology and make it easier for readers to follow the central theme and take-home messages.

(a) Move "Section 3 The Study Area" to before "Section 2 Methodology"

We prefer to keep the methodology section before the study site one in order to put more emphasis in the genericity of our methodology, which could be applied elsewhere. Nevertheless, if the final decision is to switch those sections, this modification will be done.

(b) List all variables in a table with definitions. It is hard to remember the 10s of mathematical symbols mentioned later – one has to go back and find their definitions.

A list of all variables, their symbols and definition is proposed in Appendix D. We will make sure that this is clearly indicated and visible in the manuscript.

(c) In presenting results, please use plain English of the meaning of the variable, rather than the ratio of 2 variables defined earlier, so the reader can grasp the meaning of the results.

→ *Suggested change in manuscript:* We propose to replace the ratios by plain English in the manuscript. We also propose to remind the meaning of variables more frequently in the text.

(d) Is the lateral hydrograph, QL, the same as IGF? Both appeared in the text frequently, and it appeared that they are used in the same context. If so, please unify the terminology. If not, please make it more clear by using a conceptual model diagram.

The lateral hydrograph $Q_L$ is the association of all lateral flows occurring at the elementary catchment scale ($P_{eff}$ + IGF + $\delta$, see our response to Comment 1c).

Expressing the water balance gives: $Q_I + Q_L = Q_O$ (a)

Replacing $Q_L$ in (a) gives: $Q_I + P_{eff} + IGF + \delta = Q_O$ (b)

As $\delta$ is not measured, it cannot be differentiated from IGF: $IGF + \delta = IGF^* = Q_O - Q_I - P_{eff}$ (c)

A definition of $Q_L$ is given when this term is first cited, lines 118 to 119. It is also reminded in the legend of Figure 1.

→ *Suggested change in manuscript (Methodology section):* We propose to make the distinction between IGF and $Q_L$ clearer by introducing an explicit equation.

(e) The authors talk about losing reach vs gaining reach, in the context of losing water to other catchments via IGF. The terms "losing streams" or "gaining stream" have been understood as stream and local groundwater exchange, regardless of the source/sink is local or remote (IGF). Perhaps use "losing catchment" vs "gaining catchment" because here the authors refer to IGF?

Thank you, we agree with this remark and will implement it to the manuscript.

(f) in Fig 1, catchments include headwater, intermediate, but is it intermediate between a headwater and a tailwater catchment? Is there another catchment below the intermediate?

Our spatial scale of work is the elementary catchment, defined as the drained area between two gauging stations. In case of a headwater catchment, the elementary catchment corresponds to the whole topographic catchment, as there is no gauging station to delineate the upstream divide. In other cases, the elementary catchments are intermediate catchments (with upstream and downstream gauging stations), knowing that a basin could have a succession of intermediate catchment if there are several stations on the river.

In Figure 1, there is another intermediate catchment below the presented one. We do not use the term tailwater catchment, as there are other gauging stations downstream of our study zones.

→ *Suggested change in manuscript (Introduction & Figure 1):* We propose to make Figure 1 clearer by adding an "intermediate catchment" label to the third catchment, and an "elementary catchment" label for all three catchments. See also suggestions when defining elementary catchments in the Introduction section (response to comment 1b)

Line 296. Is this stream loss, or simply that the infiltration over the surface catchment drained elsewhere, to neighboring catchments? IBF is not just a result of stream flow loss; infiltrated water may not go to the streams at all in its own catchment but may enter streams in other catchments.

We agree on the fact that IGF can be the result of both streamflow losses and surface infiltration. Nevertheless, the terms cited in line 296 ($V_S$ and $V_U$) only regard streamflow losses. Indeed, they are obtained by difference of inlet and outlet streamflow, regardless of precipitations. On the opposite side, the parameter IGF* can be interpreted by local streamflow losses or diffuse infiltration.

Cited literature:

Bailly-Comte, V., Jourde, H., Pistre, S., 2009. Conceptualization and classification of groundwater–surface water hydrodynamic interactions in karst watersheds: Case of the karst watershed of the Coulazou River (Southern France). Journal of Hydrology 376, 456–462. https://doi.org/10.1016/j.jhydrol.2009.07.053

Bonacci, O., Ljubenkov, I., Roje-Bonacci, T., 2006. Karst flash floods: an example from the Dinaric karst (Croatia). Natural Hazards and Earth System Science 6, 195–203.

Charlier, J.B., Moussa, R., Bailly-Comte, V., Danneville, L., Desprats, J.F., Ladouche, B., Marchandise, A., 2015. Use of a flood-routing model to assess lateral flows in a karstic stream: implications to the hydrogeological functioning of the Grands Causses area (Tarn River, Southern France). Environmental Earth Sciences 74, 7605–7616.

Charlier, J.B., Moussa, R., David, P., Desprats, J.F., 2019. Quantifying peakflow attenuation/amplification in a karst river using the diffusive wave model with lateral flow. Hydrological Processes. https://doi.org/10.1002/hyp.13472

De Waele, J., Martina, M.L.V., Sanna, L., Cabras, S., Cossu, Q.A., 2010. Flash flood hydrology in karstic terrain: Flumineddu Canyon, central-east Sardinia. Geomorphology 120, 162–173. https://doi.org/10.1016/j.geomorph.2010.03.021

Delrieu, G., Nicol, J., Yates, E., Kirstetter, P.-E., Creutin, J.-D., Anquetin, S., Obled, C., Saulnier, G.-M., Ducrocq, V., Gaume, E., 2005. The catastrophic flash-flood event of 8–9 September 2002 in the Gard Region, France: a first case study for the Cévennes–Vivarais Mediterranean Hydrometeorological Observatory. Journal of Hydrometeorology 6, 34–52.

López-Chicano, M., Calvache, M.L., Martín-Rosales, W., Gisbert, J., 2002. Conditioning factors in flooding of karstic poljes—the case of the Zafarraya polje (South Spain). CATENA 49, 331–352. https://doi.org/10.1016/S0341-8162(02)00053-X

Maréchal, J.C., Ladouche, B., Dörfliger, N., 2008. Karst flash flooding in a Mediterranean karst, the example of Fontaine de Nîmes. Engineering Geology 99, 138–146. https://doi.org/10.1016/j.enggeo.2007.11.013

Zanon, F., Borga, M., Zoccatelli, D., Marchi, L., Gaume, E., Bonnifait, L., Delrieu, G., 2010. Hydrological analysis of a flash flood across a climatic and geologic gradient: The September 18, 2007 event in Western Slovenia. Journal of Hydrology 394, 182–197. https://doi.org/10.1016/j.jhydrol.2010.08.020

---

## Author Comment (AC2) · 9 Sep 2020

**Response to Anonymous Referee #2**
**(Comment received and published 3 August 2020)**

We are very grateful for the constructive comments of anonymous referee #2 on the manuscript. We believe that they will help increasing substantially the scientific quality of the manuscript. We agree with most recommendations and we give in this note our responses to all remarks.

**Comment 2a)**

The authors present an interesting study about karst characteristics of catchments at storm-event timescale. The merit of the study is the treatment of multiple sites and the storm-event time-scale, where the case study region in France provides a good example and good data.

Thank you.

The overall article is well written and of scientific interest, but the scientific argumentation should be improved. The literature study is very short and lacking some other approaches, the research question is not clearly stated, the concept is not sufficiently brought into the context of other methods, and the benefits and shortcomings of the developed method against other methods should be better elaborated in the theoretical part, and later on discussed based on the results.

Thank you. We propose to modify the manuscript according to your remarks as detailed below. Your review will allow several improvements such as a more complete literature review making more understandable the originalities of the study, and a better justification of our choice to work with topographic catchments as a spatial reference.

**Comment 2b)**

In the literature study, other concepts for describing the hydrological characteristics of larger scale karst systems should be mentioned. Gárfias-Soliz et al. (2010) found system memory, response time and mean delay between input and output as indicators for karstification but emphasized that the physically founded regionalization needs to account for structural complexity, heterogeneity of the lithology and the degree of karstification. Hartmann et al. (2013) described the responsive behavior of large-scale karst systems by system signatures derived from hydrodynamic and hydrochemical observations based on data from five different karst systems. Basha (2020) presented six recession curves for the classification of karst aquifers, describing the dichotomy of the dual flow characteristics between the fissured matrix and the conduit system. How do the descriptors found in this study relate to the indicators and signatures found by other authors? Is there a clear difference between daily and storm-event resolution at the size of catchment under consideration?

Our literature study mostly highlights lacks of research on karst hydrology when the studied object is the stream (rather than the spring). We agree that it can be improved by including some references to existing works related to the developed methodology, such as some relevant articles proposed. To this end, we propose to reshape the introduction and include the following paragraphs (see also our response to referee #1 for other literature review improvements).

*→ Suggested change in manuscript (Introduction section):  The diversity of observed processes during storm event in karst catchments does not allow drawing a straightforward analysis on the control of karst in flood runoff generation. In the purpose of understanding the mechanisms involved in this control, there is a need for regionalized studies, covering a large-scale analysis of karst impact over short time periods when the catchment reacts after storm events. It is reasonable to think that karst can alternately increase or decrease storm impacts, depending on its capacity to infiltrate precipitation or to release stored water, i.e. depending on the direction of IGF it promotes. Despite the early conceptualization of IGF (Eakin, 1966), its major role in karst hydrological processes is tackled by very few studies (Le Moine et al., 2007; Lebecherel et al., 2013). Some authors tried to improve model capacities to reproduce karst-based IGF, such as Nguyen et al. (2020) with SWAT, Le Moine et al. (2008) with GR4J or Scanlon et al. (2003) comparing a distributed and a lumped model. Nevertheless, those studies dedicated to the improvement of model performance are not devoted to describe and understand all flood components in karst catchments.*

*On one hand, most studies including karst system descriptors are based on a purely hydrogeological point of view, and are very integrative, as they tend to characterize karst aquifer as a whole, by analysing daily spring discharge. Gárfias-Soliz et al. (2010) fond that system memory, response time and mean input-output delay are relevant indicators for karstification, in addition to a necessary consideration of the structural complexity and heterogeneity of the lithology. Hartmann et al. (2013), using 10 system signatures, performed a model parameters sensitivity analysis to investigate their links with hydrological processes on five Europe and Middle East karst sites. Basha et al. (2020) proposed six recession curve equations for the classification of karst aquifers, depending on their flow characteristics. On the other hand, spatialized studies mostly focus on low-flow issues and surface-water/groundwater interaction (e.g., Covino et al., 2011, Mallard et al., 2014). Moreover, most regionalization studies tend to spatialize annual indices (Sivapalan et al., 2011) or model parameters (Parajka et al., 2005; Oudin et al., 2008), and usually exclude catchments with identified IGF (Merz and Blöcshl, 2004) or karst areas (e.g., Laaha and Blöschl, 2006).*

**Comment 2c)**

"Despite some studies describing significant IGF in karst areas (Le Moine et al., 2007; Lebecherel et al., 2013), the specific issue of IGF in karst has not been addressed as such."  There have been studies dealing with IGF, in particular the modelling of IGF at catchment scale or adding IGF to hydrological catchment models. Even if karst-based IGF  is  often  neglected,  it  found  attention  among  hydrological  modelers,  introducing or enhancing karst capabilities of e.g.  SWAT (e.g., Nguyen et al., 2020).  Apart from modelling and data analyses, geo-chemical methods based on analysis of chlorides and stable isotopes has been used to characterize karst systems.

Thank you for this constructive comment. We will include a discussion on model improvements accounting for karst-based IGFs, such as the already cited work of Le Moine et al. (2008) and the study of Nguyen et al. (2020). Is also of interest the works of Scanlon et al. (2003) which modelled IGF in a karst aquifer using an equivalent porous media.

*→ Suggested change in manuscript (Introduction & discussion sections):* Please refer to our response to comment 2b for the proposed modifications.

The authors correctly mention that many hydrological studies neglect the characteristics of karst regions, but their own catchment delineation follows the "classical" orographic approach using surface water divides instead of trying to derive subsurface catchments based on hydrogeological data/maps. This is a major shortcoming of the method. In particular, the water balance descriptors could be more meaningful if the overall contributing area including the subsurface catchment would be regarded. Whenever catchment sizes and rainfall amounts are related with stream flow, there would be a mismatch in karst regions and it is unclear for me how the authors have regarded that aspect. A detailed discussion is missing. In particular, for the description of storm peak flow, I would expect water balance descriptors, which include the areas connected by IGF. For arid basins, Wolaver et al. (2008) developed a method to delineate karstic aquifers based on data. Applying such delineation beyond the usage of surface catchments might help to improve the results of the characterization of the flow response to storm events. Here, one could think that the method is inconsistent and I ask the authors to explain why they did not try to delineate the karstic aquifers and include them into the water balance considerations.

We fully agree on the fact that the hydrogeological catchment should be considered in groundwater studies. Yet, we draw attention on the fact that our spatial reference scale is not the aquifer, but the river reach (at a gauging station or at a reach delimited by two stations). For this reason, and as our study accounts for IGF, the topographic catchment appears to be the right reference, even in the case of karst catchments. This comment 2d) is very similar to the comment 1b) of Referee #1, thus we invite readers to see our response to this last comment. We propose to modify Figure 1 and to add the following paragraph to our introduction for a better understanding of our strategy.

→ *Suggested change in manuscript (Introduction section): Regional spatial analyses need to be based on reliable data at the highest resolution available. For this purpose, the scale of the elementary catchment - i.e. subdivision of a basin following available gauging stations – appears to be the best resolution for long-term monitoring. Elementary catchment can be either the drained area of a headwater catchment controlled by a gauging station, or the drained area between two gauging stations (intermediate catchment). When considering surface and groundwater components, the delineation method of elementary catchments is questionable (topographic vs. hydrogeological boundaries). Despite the importance of groundwater processes in karst areas, topographic catchment delineation remains a more robust reference, for several methodological reasons. First, IGF can be defined as groundwater flow crossing topographic divides, as this concept emerged with the evidence of certain groundwater systems extending beyond the limits of valleys (Eakin, 1966). A perfectly delineated groundwater basin would then show IGF equal to zero. For this reason, studies related to IGF use the topographic catchment spatial reference (Genereux et al., 2005; Schaller and Fan, 2009; Bouaziz et al., 2018; Nguyen et al., 2020; see also a synthesis in Fan, 2019). Second, although in karst catchments groundwater contributes to flood flow, surface runoff has to be considered as an important component of flood flow. Consideration of hydrogeological catchments could thus lead to wrong surface contribution assessment depending on their surface drainage network. Third, as some groundwater flows are aligned with the main surface drainage axis, hydrogeological catchments would encompass the whole river, making it impossible to study the spatial variability of parameters along the river, at the elementary catchment scale. Finally, topographic delineation is reliable and easily reproducible, while groundwater delineation is characterized by a strong uncertainty and variability in karst areas.*

→ *Suggested change in manuscript (Figure 1):* Add schematic delineations of topographical and groundwater basins.

The conclusions seem to ignore the efforts done in karst research over the last decades: "Existence of karst hydrological specificities has been known for decades, but is poorly quantified and documented.". Here, I disagree. The authors should explicitly say, where their method provided methodological innovation and which characteristics of karst regions could be now better described.

This statement refers to large scale works focusing on hydrological processes at the stream scale. Our purpose was not to ignore numerous works carried out in karst hydrogeology. However, the stream/river scale is not as much documented as the spring one. We agree that the text can be improved, especially in the light of the added references to advances in karst hydrology.

→ *Suggested change in manuscript (Conclusion section):* We propose to replace the cited sentence by a brief summary of what are the innovations of our study compared to existing works (storm-event time scale, spatialized quantification of karst impact on river reaches, effects on flood processes).

Literature cited:

Basha, H.A.: Flow Recession Equations for Karst Systems. Water Resources Research 56(7), 2020, Article number e2020WR027384

Gárfias-Soliz, J., Llanos-Acebo, H., Martel, R.: Time series and stochastic analyses to study the hydrodynamic characteristics of karstic aquifers. Hydrological Processes24(3), January 2010, Pages 300-316.

Hartmann, A. et al.: Process-based karst modelling to relate hydrodynamic and hydro-chemical characteristics to system properties. Hydrology and Earth System Sciences17(8), 2013, Pages 3505-3521.

Nguyen, V.T., Dietrich, J., Uniyal, B.: Modeling interbasin groundwater flow in karst areas: Model development, application, and calibration strategy. Environmental Modelling and Software 124, 2020, Article number 104606.

Wolaver, B.D. et al.: Delineation of regional arid karstic aquifers: An integrative data approach. Ground Water 46(3), 2008, Pages 396-413.

Anaya, R., Wanakule, N., 1993. A Lumped Parameter Model for the Edwards Aquifer (Technical Report). Texas Water Resources Institute.

Barrett, M.E., Charbeneau, R.J., 1997. A parsimonious model for simulating flow in a karst aquifer. Journal of Hydrology 196, 47–65. https://doi.org/10.1016/S0022-1694(96)03339-2

Bouaziz, L., Weerts, A., Schellekens, J., Sprokkereef, E., Stam, J., Savenije, H., Hrachowitz, M., 2018. Redressing the balance: quantifying net intercatchment groundwater flows. Hydrology and Earth System Sciences 22, 6415–6434. https://doi.org/10.5194/hess-22-6415-2018

Covino, T., McGlynn, B., Mallard, J., 2011. Stream-groundwater exchange and hydrologic turnover at the network scale: Hydrologic turnover at the network scale. Water Resources Research 47. https://doi.org/10.1029/2011WR010942

Eakin, T.E., 1966. A regional interbasin groundwater system in the White River Area, southeastern Nevada. Water Resources Research 2, 251–271. https://doi.org/10.1029/WR002i002p00251

Fan, Y., 2019. Are catchments leaky? Wiley Interdisciplinary Reviews: Water 6. https://doi.org/10.1002/wat2.1386

Genereux, D.P., Jordan, M.T., Carbonell, D., 2005. A paired-watershed budget study to quantify interbasin groundwater flow in a lowland rain forest, Costa Rica. Water Resources Research 41. https://doi.org/10.1029/2004WR003635

Hartmann, A., Barberá, J.A., Lange, J., Andreo, B., Weiler, M., 2013. Progress in the hydrologic simulation of time variant recharge areas of karst systems – Exemplified at a karst spring in Southern Spain. Advances in Water Resources 54, 149–160. https://doi.org/10.1016/j.advwatres.2013.01.010

Laaha, G., Blöschl, G., 2006. Seasonality indices for regionalizing low flows. Hydrological Processes 20, 3851–3878. https://doi.org/10.1002/hyp.6161

Le Moine, N., Andréassian, V., Mathevet, T., 2008. Confronting surface- and groundwater balances on the La Rochefoucauld-Touvre karstic system (Charente, France): CLOSING THE CATCHMENT-SCALE WATER BALANCE-A CASE STUDY. Water Resources Research 44. https://doi.org/10.1029/2007WR005984

Le Moine, N., Andréassian, V., Perrin, C., Michel, C., 2007. How can rainfall-runoff models handle intercatchment groundwater flows? Theoretical study based on 1040 French catchments: Dealing with IGF in rainfall-runoff mode. Water Resources Research 43 . https://doi.org/10.1029/2006WR005608

Lebecherel, L., Andréassian, V., Perrin, C., 2013. On regionalizing the Turc-Mezentsev water balance formula. Water Resources Research 49, 7508–7517. https://doi.org/10.1002/2013WR013575

Mallard, J., McGlynn, B., Covino, T., 2014. Lateral inflows, stream-groundwater exchange, and network geometry influence stream water composition. Water Resources Research 50, 4603–4623. https://doi.org/10.1002/2013WR014944

Merz, R., Blöschl, G., 2004. Regionalisation of catchment model parameters. Journal of Hydrology 287, 95–123. https://doi.org/10.1016/j.jhydrol.2003.09.028

Oudin, L., Andréassian, V., Perrin, C., Michel, C., Le Moine, N., 2008. Spatial proximity, physical similarity, regression and ungauged catchments: A comparison of regionalization approaches based on 913 French catchments. Water Resources Research 44. https://doi.org/10.1029/2007WR006240

Parajka, J., Merz, R., Blöschl, G., 2005. A comparison of regionalisation methods for catchment model parameters. Hydrology and Earth System Sciences 16.

Scanlon, B.R., Mace, R.E., Barrett, M.E., Smith, B., 2003. Can we simulate regional groundwater flow in a karst system using equivalent porous media models? Case study, Barton Springs Edwards aquifer, USA. Journal of Hydrology 276, 137–158.

Schaller, M.F., Fan, Y., 2009. River basins as groundwater exporters and importers: Implications for water cycle and climate modeling. Journal of Geophysical Research 114. https://doi.org/10.1029/2008JD010636

Sivapalan, M., Yaeger, M.A., Harman, C.J., Xu, X., Troch, P.A., 2011. Functional model of water balance variability at the catchment scale: 1. Evidence of hydrologic similarity and space-time symmetry. Water Resources Research 47. https://doi.org/10.1029/2010WR009568

---

## Author Response (AR1)

Dear Editor,

Thank you for your response to our revision proposals. We appreciate your constructive comments on the manuscript and believe they will help improving the quality and understandability of the proposed article. You will find in this letter:

- I) a detailed reply to your report, with explanations on how we have updated our manuscript;
- II) the responses to reviewers comments, including the associated changes to the manuscript;
- III) the revised manuscript with revision marks displayed.

We made our best for providing modifications that fulfill the core comments and expectations of the review process, and hope that you will find this new version acceptable for publication.

Martin Le Mesnil, on behalf the co-authors

*NB: In the present document, line numbers refer to the revised manuscript with all revision marks displayed, as attached at the end of this file.*

**I)     Response to Editor**

Comments to the Author:
The reviewers are generally favourable to the paper and have made some useful comments that will improve the manuscript, however they each note the need for major corrections. The core matter, and I agree, is a lack of more detailed analyses as to the components of the fluxes and the different assumptions that are made in that quantification to gain, for example IGF values.

We agree that some more detailed analyses are required to improve the manuscript, notably on the question of IGF values and the relative processes than control it. Therefore, we added to section 5.3 the following figure and discussion that tackles the question of differentiated IGF impacts on the quick- and slow-flow components.

→ *Performed change in manuscript (Discussion section Line 570 to Line 581 in the revised version of the manuscript and new Figure 11):*

[Figure]

*Figure 11 : VS depths versus IGF\* depths for all events and reaches, with differentiation of geology types*

*"Figure 11 represents $V_S$ versus IGF\* (both expressed in depth) for all events and reaches, with different symbols according to geology type. $V_S$ being equal to $V_{S,O} - V_{S,I}$, the $V_S$ versus IGF\* relationship provides an insight into the relative proportions of total IGF\* depth and quick-flow component variation. First, while IGF\* values vary from -200 to 300 mm, $V_S$ values are mostly*

*positive, ranging from -20 to 150 mm. This indicates that for most events, when outgoing IGFs (negative IGF\* values) occur, the quick-flow component still increases. This is particularly true on NK catchments, whereas some events with negative IGF\* on K catchments show negative (or low positive) $V_S$ values. In that case, a likely hypothesis is the presence of losses in karst riverbeds via sinkholes, which are not present in non-karst reaches (see such example in Charlier et al., 2019). Regarding events with positive IGF\*, $V_S$ depths are mainly inferior to IGF\* ones, with values around half of the total IGF\* ones. This shows that incoming IGFs during storm-events feed the slow-flow component of the total streamflow in similar proportions than the quick-flow component. This is linked to the different IGF processes occurring at the catchment scale: slow-flow component corresponding to aquifer drainage by the river (e.g. Bailly-Comte et al., 2012) and quick-flow component corresponding to karst spring activation (Bonacci and Bojanic, 1991; Maréchal et al., 2008), for example."*

**Comment 2)**

I fully appreciate what the authors have written in their response in terms of clarifying the choices made in additional text (i.e. both reviewers commented on the need for more transparency and critical analyses on the use of a topographic catchment basin).

Thank you.

**Comment 3)**

Whilst I appreciate the authors current responses and suggested changes I don't think yet they fully tackle the issues of being more transparent on how the flow components come together per reach. What that looks like more clearly in the results for each reach and if those numbers seem physically plausible given the assumptions that need to be made. We have to remember here the core analyses are conducted on event behaviour and there are some challenging questions in karstic landscapes how lags, and pre-event events may indeed impact on the expected current storm event contributions, as well as seasonal variations that might well be likely for different 'general' aquifer states. I encourage and expect therefore the authors to try to tease out these issues more (i.e. the need for improve critical analyses and new results) to make the results and how they come together are more transparent, and a little more of a critical analyses and assumptions required to be identified.

As you mention, other factors control catchments flood response, such as seasonal parameters. We added the following figure and discussion to section 5.1, that analyses the seasonal effect and investigated possible causes of this variability.

→ *Performed change in manuscript (Discussion section Line 479 to Line 492 in the revised version of the manuscript and new Figure 8):*

[Figure]

*Figure 8 : IGF\* depths distribution for the three studied sites, according to seasons*

*"Hydrological processes have been shown to be influenced by physiographic parameters such as karst occurrence. Nevertheless, other drivers can control catchments hydrological response to storm events. As an example, figure 8 shows IGF\* depths distribution for the three studied sites, according to seasons. Seasons have been selected in order to reflect the main periods of the hydrological year, and to be suitable for the three sites. Throughout the hydrological year (from September to August), median IGF\* depth is continuously decreasing, for all sites towards zero. This is probably linked to the hydrological conditions of catchments, in particular the saturation state of aquifers. Indeed, low water table periods (Apr to Aug) are more likely to limit IGFs. In the case of Jura catchments, a majority of events show IGF gains during high-flow periods (fall and winter), whereas a majority of events promote IGF losses during low-flow periods. This reversal of IGF direction, in the particular case of karst catchments, can also be favoured by the complex organization of the undergournd conduit networks. In fact, connection or disconnection of the main networks could temporally modify the hydrogeolocical catchment boundaries with threshold effects (Charlier et al., 2012; Bonacci, 2015). For those reasons, spatial and temporal variability of karst catchments behaviour are still challenging to characterize and predict. A comprehensive understanding of flood processes would thus imply accurate and continuous monitoring of climatic, hydrological and hydrogeological variables, on multiple catchments covering various physiographical settings."*

**Comment 4)**

Plus also other ways to help the reader better interpret the data provided. i.e. if the IGF is in mm, then what fraction of the catchment would be increased or lost to effectively result in that mm gain for the hydrogeologically active component (for example) - I think this would be really interesting to look at to see if the differences are plausible under a certain set of assumptions. Or indeed what fraction of the storm flow is IGF? Even examples of the flow components for some reaches, perhaps even reported in Supplementary pages?

Thank you for these remarks. As mentioned in Comment 1, we propose to analyse which part of quick- and slow-flow components is affected by IGF. Moreover, we agree that the estimation of the catchment fraction to be added or removed to result in the IGF gain or loss, is a good way to make the results more interpretable and visual. Therefore, we added the following figure and discussion to section 5.2. It allows analysing the differences between topographic and "hydrogeologically active" catchment areas, and discuss their consistency.

→ *Performed change in manuscript (Discussion section Line 523 to Line 539 in the revised version of the manuscript and new Figure 9):*

[Figure]

Figure 9 : Main geological features in the three studied areas (a, b, c), median storm event IGF* values (including potential aquifer storage variation) (d, e, f), and area variation from actual topographic to virtual hydrogeologically active (in %) (g, h, i) and in km² (j, k, l).

*"Figures 9g to 9i show the percentage of area variation of the elementary catchments, from the topographic catchment area to the fictive area of the hydrogeologically active catchment (i.e. without excess or deficit in water balance). The area of such catchments is calculated from the IGF\* and $V_P$ depths. Assuming a spatially homogeneous rainfall intensity, the area variation corresponds to the surface to be added or withdrawn, in regards to the topographic elementary catchment, to obtain the median IGF\* value. These variations reflect the incoming or outgoing IGF processes, respectively. For Cévennes and Jura catchments, the values of area variation are of similar magnitudes, ranging from approximately -7 % to +90 %. Regarding Normandy catchments, area variations are slightly lower, with values ranging from -20 % to +32 %. This shows that IGF can be an important term of the water balance during storm events, and has to be considered as such, in order to better understand flow processes during floods.*

*Figures 9j to 9l show the same elementary catchments area variation, expressed in km². These values are correlated to the size of topographic catchments. They allow a concrete representation of the recharge areas located outside of the surface catchments. As this study is based on storm-events only, and considering that the studied catchments extend further downstream, the sum of fictive active catchments (which is a first approximation of the hydrogeological catchment) do not match with the topographic catchment area. Nevertheless, a previous study by Le Mesnil et al. (2020) showed that, along the Doubs river, annual IGF take a part in the water balance that progressively decreases from spring to outlet and tends towards zero."*

So I'd like to see a bit more in the changes to the paper than what has been specified by the authors in their response to this point and I believe it will make it a better paper...., I shall then conduct an editorial review of the changes, but if need be I will send this back for further review if I don't think the spirit of what the reviewers have noted is being further developed by the authors, best wishes, Jim

Thank you for your decision, we hope that our modifications will meet your expectations.

**Response to Anonymous Referee #1**

We are very grateful for the constructive comments of anonymous referee #1 on the manuscript. We believe that they will help increasing substantially the scientific quality of the manuscript. We agree with most recommendations and we give in this note our responses to all remarks.

**Comment 1a)**

The study analyzed stream flow observations at 108 gages in three regions with varying karst area, focusing on event-scale hydrologic response to 20 large storm events. Using 15 descriptors on catchment water balance, hydrograph shape, and difference between an upstream and downstream gage, the study compared the catchment response in karst (K), middle (M) and non-karst (NK) catchments.  It is concluded that (1) karst promotes high infiltration but (2) slows down flood response (both rising and falling limb), and the latter behavior is attributed to inter-basin groundwater flow (IGF).

I find the study valuable in that it selected a wide range of catchments under different climatic and geologic conditions, and that it focused on event time-scale response of catchment runoff underlain by karst geology.   However, the manuscript can benefit significantly from a clearer and sounder conceptual model, better framing of the questions (what we know and don't know) and the hypotheses to be tested with this particular dataset, how the results speak for or against the hypotheses posed, and a better overall presentation, as expanded below.

We modified the manuscript according to remarks of referee #1 as detailed below. This review will allow several improvements such as a better justification of our choice to work with topographic catchments as a spatial reference, a more complete literature review, and a clearer definition of what is the lateral flow $Q_L$ in comparison to IGF.

**Comment 1b)**

First, there are some conceptually ambiguities.

(a) My understanding is that groundwater flow, particularly trans-catchment groundwater flow, or inter-basin flow (IBF), does not follow surface drainage structure in karst terrain as depicted in Figure 1.  We cannot delineate groundwater basins based on surface topography, a well-known problem in karst terrain. Figure 1 depicts the IBF as completely defined by, and in parallel with the surface drainage gradient.  While this may be fine for unconsolidated materials without strong geologic structure, it is hardly the case for karst terrain, where the extensive underground conduit networks do not follow the topography.  The authors' finding

that the strongest lateral inflow is in inter-mediate catchments in NK, that is, there is more IBF in non-karst areas, is counterintuitive and confusing, and points to this conceptual flaw. The reason may well be that in NK areas the surface and subsurface drainage system are more aligned, and such an assumption is more valid, allowing the detection of IBG inflow. The authors need to clarify and justify this assumption, because it also underlies the methodology used in the study, with the analyses entirely based on streamflow.

First, the term IGF was kept in the manuscript (instead of IBF) as it is a term dedicated to the study of Interbasin Groundwater Flow.

Figure 1 represents all identified inputs and outputs at the scale of an elementary catchment (drained area between two gauging stations). They are composed of three main flows: input streamflow $Q_I$, output streamflow $Q_O$ and lateral flow $Q_L$. $Q_L$ is composed of several flows corresponding to different processes, which are numbered represented on Figure 1. Interbasin Groundwater Flow (IGF) is represented by incoming and outgoing arrows of two types. Straight arrows represent IGF in parallel with the surface drainage gradient, while curved arrows represent IGF from/towards other directions (outside upstream or downstream elementary catchments).

→ *Performed change in manuscript (Figure 1):* Modification of IGF arrows to make clearer the fact that IGF can be in different directions than the main surface drainage direction, and to differentiate them from other $Q_L$ flows.

We fully agree on the comment stated that groundwater basins cannot be delineated based on surface topography, especially in karst areas. However, our spatial reference is the stream catchment for which the topographic catchment is the relevant spatial reference, even when accounting for groundwater flows in addition to surface flows. We added the following paragraph to our introduction for a better understanding of our strategy.

→ *Performed change in manuscript (Introduction section Line 72 to Line 88 in the revised version of the manuscript): "Regional spatial analyses need to be based on reliable data at the highest resolution available. For this purpose, the scale of the elementary catchment - i.e. subdivision of a basin following available gauging stations – appears to be the best resolution for long-term monitoring. Elementary catchment can be either the drained area of a headwater catchment controlled by a gauging station, or the drained area between two gauging stations (intermediate catchment). When considering surface and groundwater components, the delineation method of elementary catchments is questionable (topographic vs. hydrogeological boundaries). Despite the importance of groundwater processes in karst areas, topographic catchment delineation remains a more robust reference, for several methodological reasons. First, IGF can be defined as groundwater flow crossing topographic divides, as this concept emerged with the evidence of certain groundwater systems extending beyond the limits of valleys (Eakin, 1966). A perfectly delineated groundwater basin would then show IGF equal to zero. For this reason, studies related to IGF use the topographic catchment spatial reference (Genereux et al., 2005; Schaller and Fan, 2009; Bouaziz et al., 2018; Nguyen et al., 2020; see also a synthesis in Fan, 2019). Second, although in karst catchments groundwater contributes to flood flow, surface runoff has to be considered obviously as an important component of flood flow. Consideration of hydrogeological catchments could thus lead to wrong surface contribution assessment depending on their surface drainage network. Third, as some groundwater flows are aligned with the main surface drainage axis,*

*hydrogeological catchments would encompass the whole river, making it impossible to study the spatial variability of parameters along the river, at the elementary catchment scale. Finally, topographic delineation is reliable and easily reproducible, while groundwater delineation is characterized by a strong uncertainty and variability in karst areas."*

**Comment 1c)**

(b) The methodology infers IBF into and out of a catchment by comparing the inflow QI and outflow Qo of a stream reach (Fig 1). The authors need to clarify to what extend one can infer IGFs from streamflow alone.

Considering the topographic elementary catchment as our reference for all water balance calculations, IGF is inferred from inflow $Q_I$, outflow $Q_o$ and effective rainfall $P_{eff}$ (obtained by subtracting the estimated evapotranspiration to measured rainfall, see Appendix C of the submitted manuscript). Knowing that this term also accounts for potential aquifer level variations, it is noted IGF*.

The difference reflects inflow from local precipitation falling on its topographic catchment and subsequent infiltration and surface and subsurface runoff (local source), the inflow from upstream catchments via IBF (remote source), minus the loss to local aquifers (local sink, recharging local aquifers, increasing groundwater storage) which may or may not leave the catchment via IBF (remote sink). But without an explicit aquifer water balance to track all the terms during the events, it is difficult to separate these terms. To arrive at quantitative results, the authors need to explicitly quantify these terms, incorporating aquifer water level observations, spring discharges.

As explained in lines 162 to 166 and visible in Figure 1, lateral exchanges may be a combination of (supposed decreasing importance):

- Effective rainfall (precipitation minus evapotranspiration) over the elementary catchment,
- IGF,
- Aquifer storage variation,
- Overbank phenomena.

We consider overbank flow to be negligible, assuming that the overflow water returns to the river after a relatively short time during the recession. We propose an assessment of evapotranspiration (explained in Appendix C) which allows estimating the effective rainfall on the elementary catchments. Thus, combined to inlet streamflow $Q_I$ and outlet streamflow $Q_O$ data, our methodology allows calculating the remaining term of the water balance, noted IGF*, which includes IGF and the potential aquifer storage variation during the considered event (noted δ). δ can either be positive, corresponding to an aquifer recharge, or negative, corresponding to an aquifer draining. Aquifer draining is unlikely during storm events, as important rainfalls generally occur. In case of an aquifer recharge, and as our analysis is performed on the whole storm-event period (including the entire recession), a substantial part

of the infiltrated water should be released. For these reasons, δ is assumed to be a minor component on the IGF* term compared to IGF. This is discussed in lines 182 to 195, and calculation of IGF* is presented in equation 8.

**→ Performed change in manuscript (Methodology section Line 192 in the revised version of the manuscript):** introduction of equation 6: "$Q_L = IGF + Peff + δ$"

The diffusive wave model for surface water propagation appears to be an over-kill in comparison to a complete lack of first-order water balance in the subsurface.

We precise that the diffusive wave model is used to analyse the flow dynamics during floods through the simulation of lateral hydrographs. It is not used in the lateral flow estimation for water balance calculations (see our response above).

**Comment 1d)**

Second, the authors should provide a more thorough literature review, discuss what we know and do not know in terms of event-scale catchment runoff generation in karst terrain, and pose a set of testable hypotheses. For example, it is expected that in karst area, infiltration is high, and storm flow is flashy due to the open conduits in the subsurface, but after a threshold when the groundwater builds up to the spring overflow point. Then as the results are opposite (slow hydrograph response), one can reason why this hypothesis must be rejected. For another example, one can hypothesize that in karst catchments, streamflow can be lower or higher than expected from calculations based on infiltration over the surface catchment area alone (local source), or there is mass imbalance, and thus IGF must be invoked. Then the analyses can be targeted to test these hypotheses, and the results discussed with clarity surrounding these hypotheses.

We developed the literature review (see also our response to referee #2 regarding this aspect), and especially discussed in a larger extent what are the main findings on karst impacts on flood processes and runoff generation. We included the following paragraph to the introduction. This more detailed analysis was also used to better interpret and discuss our results.

**→ Performed change in manuscript (Introduction section Line 36 to Line 44 in the revised version of the manuscript):** "Karst impacts on flood processes are mostly documented through case studies. As an example, Zanon et al. (2010) showed that during a flash flood in 2007 in Slovenia, karst area reduced flooding, which was more important in a non-karst neighbouring zone, receiving less precipitation. Likewise, Delrieu et al. (2005) observed, for an exceptional storm event in 2002, lower runoff coefficient values for the karstic catchment compared to the hard-rock catchment in the eastern zone of the Cévennes Mountains. De Waele et al. (2010) and Charlier et al. (2015, 2019) determined that, depending on the location on the river profile, karst areas could result in streamflow losses or gains due to the high spatial variability of the hydrogeological karst features. Other frequently described processes are groundwater rising leading to reduced infiltration and important surface runoff (Lopez-Chicano et al., 2002; Bonacci et al., 2006), and backflooding/sinkhole flooding due to conduit constriction (Maréchal et al., 2008; Bailly-Comte et al., 2009)."

Third, in addition to the above, a few other things can be done to reduce the length, enhance focus, clarify terminology and make it easier for readers to follow the central theme and take-home messages.

(a) Move "Section 3 The Study Area" to before "Section 2 Methodology"

We prefer to keep the methodology section before the study site one in order to put more emphasis in the genericity of our methodology, which could be applied elsewhere.

(b) List all variables in a table with definitions. It is hard to remember the 10s of mathematical symbols mentioned later – one has to go back and find their definitions.

A list of all variables, their symbols and definition is proposed in Appendix D.

(c) In presenting results, please use plain English of the meaning of the variable, rather than the ratio of 2 variables defined earlier, so the reader can grasp the meaning of the results.

→ *Performed change in manuscript:* We replaced the ratios by plain English in the manuscript. We also reminded the meaning of variables more frequently in the text.

(d) Is the lateral hydrograph, QL, the same as IGF? Both appeared in the text frequently, and it appeared that they are used in the same context. If so, please unify the terminology. If not, please make it more clear by using a conceptual model diagram.

The lateral hydrograph $Q_L$ is the association of all lateral flows occurring at the elementary catchment scale ($P_{eff}$ + IGF + $\delta$, see our response to Comment 1c).

Expressing the water balance gives: $\qquad\qquad\qquad\qquad Q_I + Q_L = Q_O \qquad\qquad\qquad\qquad$ (a)

Replacing $Q_L$ in (a) gives: $\qquad\qquad\qquad\qquad Q_I + P_{eff} + IGF + \delta = Q_O \qquad\qquad$ (b)

As $\delta$ is not measured, it cannot be differentiated from IGF:$IGF + \delta = IGF^* = Q_O - Q_I - P_{eff}$ (c)

A definition of $Q_L$ is given when this term is first cited, lines 187 to 188. It is also reminded in the legend of Figure 1.

→ *Performed change in manuscript (Methodology section Line 192 in the revised version of the manuscript):* We made the distinction between IGF and $Q_L$ clearer by introducing an explicit equation (6): "$Q_L = IGF + Peff + \delta$"

(e) The authors talk about losing reach vs gaining reach, in the context of losing water to other catchments via IGF. The terms "losing streams" or "gaining stream" have been understood as stream and local groundwater exchange, regardless of the source/sink is local or remote (IGF). Perhaps use "losing catchment" vs "gaining catchment" because here the authors refer to IGF?

We agree with this remark that has been implemented to the manuscript.

(f) in Fig 1, catchments include headwater, intermediate, but is it intermediate between a headwater and a tailwater catchment? Is there another catchment below the intermediate?

Our spatial scale of work is the elementary catchment, defined as the drained area between two gauging stations. In case of a headwater catchment, the elementary catchment corresponds to the whole topographic catchment, as there is no gauging station to delineate the upstream divide. In other cases, the elementary catchments are intermediate catchments (with upstream and downstream gauging stations), knowing that a basin could have a succession of intermediate catchment if there are several stations on the river.

In Figure 1, there is another intermediate catchment below the presented one. We do not use the term tailwater catchment, as there are other gauging stations downstream of our study zones.

→ *Performed change in manuscript (Introduction & Figure 1):* We made Figure 1 clearer by adding an "intermediate catchment" label to the third catchment, and an "elementary catchment" label for all three catchments. See also suggestions when defining elementary catchments in the Introduction section (response to comment 1b)

Line 296. Is this stream loss, or simply that the infiltration over the surface catchment drained elsewhere, to neighboring catchments? IBF is not just a result of stream flow loss; infiltrated water may not go to the streams at all in its own catchment but may enter streams in other catchments.

We agree on the fact that IGF can be the result of both streamflow losses and surface infiltration. Nevertheless, the terms cited in line 296 [here line 368] ($V_S$ and $V_U$) only regard streamflow losses. Indeed, they are obtained by difference of inlet and outlet streamflow, regardless of precipitations. On the opposite side, the parameter IGF* can be interpreted by local streamflow losses or diffuse infiltration.

**Response to Anonymous Referee #2**

We are very grateful for the constructive comments of anonymous referee #2 on the manuscript. We believe that they will help increasing substantially the scientific quality of the manuscript. We agree with most recommendations and we give in this note our responses to all remarks.

**Comment 2a)**

The authors present an interesting study about karst characteristics of catchments at storm-event timescale. The merit of the study is the treatment of multiple sites and the storm-event time-scale, where the case study region in France provides a good example and good data.

Thank you.

The overall article is well written and of scientific interest, but the scientific argumentation should be improved. The literature study is very short and lacking some other approaches, the research question is not clearly stated, the concept is not sufficiently brought into the context of other methods, and the benefits and shortcomings of the developed method against other methods should be better elaborated in the theoretical part, and later on discussed based on the results.

We modified the manuscript according to remarks of referee #2 as detailed below. This review will allow several improvements such as a more complete literature review making more understandable the originalities of the study, and a better justification of our choice to work with topographic catchments as a spatial reference.

**Comment 2b)**

In the literature study, other concepts for describing the hydrological characteristics of larger scale karst systems should be mentioned. Gárfias-Soliz et al. (2010) found system memory, response time and mean delay between input and output as indicators for karstification but emphasized that the physically founded regionalization needs to account for structural complexity, heterogeneity of the lithology and the degree of karstification. Hartmann et al. (2013) described the responsive behavior of large-scale karst systems by system signatures derived from hydrodynamic and hydrochemical observations based on data from five different karst systems. Basha (2020) presented six recession curves for the classification of karst aquifers, describing the dichotomy of the dual flow characteristics between the fissured matrix and the conduit system. How do the descriptors found in this study relate to the indicators and signatures found by other authors? Is there a clear difference between daily and storm-event resolution at the size of catchment under consideration?

Our literature study mostly highlights lacks of research on karst hydrology when the studied object is the stream (rather than the spring). We agree that it can be improved by including some references to existing works related to the developed methodology, such as some relevant articles proposed. To this end, we reshaped the introduction and included the following paragraphs (see also our response to referee #1 for other literature review improvements).

→ *Performed change in manuscript (Introduction section Line 45 to Line 71 in the revised version of the manuscript): "The diversity of observed processes during storm event in karst catchments does not allow drawing a straightforward analysis on the control of karst in flood runoff generation. In the purpose of understanding the mechanisms involved in this control, there is a need for regionalized studies, covering a large-scale analysis of karst impact over short time periods when the catchment reacts after storm events. It is reasonable to think that karst can alternately increase or decrease storm impacts, depending on its capacity to infiltrate precipitation or to release stored water, i.e. depending on the direction of IGF it promotes. Despite the early conceptualization of IGF (Eakin, 1966), its major role in karst hydrological processes is tackled by very few studies (Le Moine et al., 2007; Lebecherel et al., 2013). Some authors tried to improve model capacities to reproduce karst-based IGF, such as Nguyen et al. (2020) with SWAT, Le Moine et al. (2008) with GR4J or Scanlon et al. (2003) comparing a distributed and a lumped model. Nevertheless, those studies dedicated to the improvement of model performance are not devoted to describe and understand all flood components in karst catchments.*

*On one hand, most studies including karst system descriptors are based on a purely hydrogeological point of view, and are very integrative, as they tend to characterize karst aquifer as a whole, by analysing daily spring discharge. Gárfias-Soliz et al. (2010) fond that system memory, response time and mean input-output delay are relevant indicators for karstification, in addition to a necessary consideration of the structural complexity and heterogeneity of the lithology. Hartmann et al. (2013), using 10 system signatures, performed a model parameters sensitivity analysis to investigate their links with hydrological processes on five Europe and Middle East karst sites. Basha et al. (2020) proposed six recession curve equations for the classification of karst aquifers, depending on their flow characteristics. On the other hand, some studies accounting for a spatialization of catchments focus on low-flow issues and surface-water/groundwater interaction (e.g., Covino et al., 2011, Mallard et al., 2014). Moreover, most regionalization works tend to spatialize annual indices (Sivapalan et al., 2011) or model parameters (Parajka et al., 2005; Oudin et al., 2008), and usually exclude catchments with identified IGF (Merz and Blöcshl, 2004) or karst areas (e.g., Laaha and Blöschl, 2006)."*

**Comment 2c)**

"Despite some studies describing significant IGF in karst areas (Le Moine et al., 2007; Lebecherel et al., 2013), the specific issue of IGF in karst has not been addressed as such." There have been studies dealing with IGF, in particular the modelling of IGF at catchment scale or adding IGF to hydrological catchment models. Even if karst-based IGF is often neglected, it found attention among hydrological modelers, introducing or enhancing karst capabilities of e.g. SWAT (e.g., Nguyen et al., 2020). Apart from modelling and data analyses, geo-chemical methods based on analysis of chlorides and stable isotopes has been used to characterize karst systems.

We included a discussion on model improvements accounting for karst-based IGFs, such as the already cited work of Le Moine et al. (2008) and the study of Nguyen et al. (2020). Is also of interest the works of Scanlon et al. (2003) which modelled IGF in a karst aquifer using an equivalent porous media.

→ *Performed change in manuscript (Introduction & discussion sections Line 51 to Line 54 in the revised version of the manuscript):* Please refer to our response to comment 2b.

**Comment 2d)**

The authors correctly mention that many hydrological studies neglect the characteristics of karst regions, but their own catchment delineation follows the "classical" orographic approach using surface water divides instead of trying to derive subsurface catchments based on hydrogeological data/maps. This is a major shortcoming of the method. In particular, the water balance descriptors could be more meaningful if the overall contributing area including the subsurface catchment would be regarded. Whenever catchment sizes and rainfall amounts are related with stream flow, there would be a mismatch in karst regions and it is unclear for me how the authors have regarded that aspect. A detailed discussion is missing. In particular, for the description of storm peak flow, I would expect water balance descriptors, which include the areas connected by IGF. For arid basins, Wolaver et al. (2008) developed a method to delineate karstic aquifers based on data. Applying such delineation beyond the usage of surface catchments might help to improve the results of the characterization of the flow response to storm events. Here, one could think that the method is inconsistent and I ask the authors to explain why they did not try to delineate the karstic aquifers and include them into the water balance considerations.

We fully agree on the fact that the hydrogeological catchment should be considered in groundwater studies. Yet, we draw attention on the fact that our spatial reference scale is not the aquifer, but the river reach (at a gauging station or at a reach delimited by two stations). For this reason, and as our study accounts for IGF, the topographic catchment appears to be the right reference, even in the case of karst catchments. This comment 2d) is very similar to the comment 1b) of Referee #1, thus we invite readers to see our response to this last comment. We added the following paragraph to our introduction for a better understanding of our strategy.

→ *Performed change in manuscript (Introduction section Line 72 to Line 88 in the revised version of the manuscript): "Regional spatial analyses need to be based on reliable data at the highest resolution available. For this purpose, the scale of the elementary catchment - i.e. subdivision of a basin following available gauging stations – appears to be the best resolution for long-term monitoring. Elementary catchment can be either the drained area of a headwater catchment controlled by a gauging station, or the drained area between two gauging stations (intermediate catchment). When considering surface and groundwater components, the delineation method of elementary catchments is questionable (topographic vs. hydrogeological boundaries). Despite the importance of groundwater processes in karst areas, topographic catchment delineation remains a more robust reference, for several methodological reasons. First, IGF can be defined as groundwater flow crossing topographic divides, as this concept emerged with the evidence of certain groundwater systems extending beyond the limits of*

*valleys (Eakin, 1966). A perfectly delineated groundwater basin would then show IGF equal to zero. For this reason, studies related to IGF use the topographic catchment spatial reference (Genereux et al., 2005; Schaller and Fan, 2009; Bouaziz et al., 2018; Nguyen et al., 2020; see also a synthesis in Fan, 2019). Second, although in karst catchments groundwater contributes to flood flow, surface runoff has to be considered obviously as an important component of flood flow. Consideration of hydrogeological catchments could thus lead to wrong surface contribution assessment depending on their surface drainage network. Third, as some groundwater flows are aligned with the main surface drainage axis, hydrogeological catchments would encompass the whole river, making it impossible to study the spatial variability of parameters along the river, at the elementary catchment scale. Finally, topographic delineation is reliable and easily reproducible, while groundwater delineation is characterized by a strong uncertainty and variability in karst areas."*

**Comment 2e)**

The conclusions seem to ignore the efforts done in karst research over the last decades: "Existence of karst hydrological specificities has been known for decades, but is poorly quantified and documented.". Here, I disagree. The authors should explicitly say, where their method provided methodological innovation and which characteristics of karst regions could be now better described.

This statement refers to large scale works focusing on hydrological processes at the stream scale. Our purpose was not to ignore numerous works carried out in karst hydrogeology. However, the stream/river scale is not as much documented as the spring one. We agree that the text can be improved, especially in the light of the added references to advances in karst hydrology. We therefore replaced the cited sentence by a brief summary of what are the innovations of our study compared to existing works (storm-event time scale, spatialized quantification of karst impact on river reaches, effects on flood processes).

→ *Performed change in manuscript (Conclusion section Line 593 to Line 598 in the new version of the manuscript):* *"
[revised manuscript text omitted]

---

## Author Response (AR3)

Dear Editor,

Thank you for your decision following reviewers' reports. We appreciate your helping and constructive implication in this review process. You will find in this letter a reply to your report and to reviewers' comments.

Best regards,

Martin Le Mesnil, on behalf the co-authors

*NB: In the present document, line numbers refer to the revised manuscript with all revision marks displayed.*

**I)   Response to Editor**

Comments to the Author:
The revised manuscript has received very favourable reviews and the authors time and effort on improving their paper is very much appreciated. I would just ask the authors to respond and reflect on Reviewer 1's comments and minor changes before publication that I shall quickly review...., Jim.

Thank you. We carefully read the reviewers comments and implemented corresponding modifications to the revised version of the manuscript.

**II)   Response to reviewers**

**Response to Anonymous Referee #2**

The authors have improved their manuscript according to the recommendations of the editor and the reviewers. In particular, they have extended their justification for using topographic catchment divides.

We are grateful for this positive feedback on the revised manuscript.

However, I think that in their description of IGF based models (l 70ff) it is not clear that conceptual models can follow a topographic delineation for surface flow components but a hydrogeologic delineation for groundwater flow components. The paragraph reads as if the concepts of topographic and hydrogeological divides would be mutually exclusive but they are not in modelling. I would personally not write that models with IGF follow a topographic delineation approach as in l 74. I agree that groundwater catchment delineation is subject to much higher uncertainty than topographic delineation.

We agree with this statement and acknowledge that models can follow a topographic delineation for surface runoff simulation in addition to a hydrogeological delineation for groundwater flow. Nevertheless, delineating hydrogeological boundaries for all our catchments would demand complementary investigations and cannot be automatically computed (unlike topographic boundaries). Moreover, those delineations would introduce important uncertainties. We modified sentences accordingly in lines 74 and 77.

I think the article can be published now with technical corrections:

- l 42 accent at Lopez missing
- l 63 "Merz and Blöschl"
- fig. 7: mixed, not mixt
- l 788: quickis -> quick is
- In some references, capitals are used. I think that the style of the list of references should be checked against the journal style.

We modified the manuscript according to these remarks.

**Response to Anonymous Referee #3**

In „Impact of karst areas on runoff generation, lateral flow and interbasin groundwater flow at the storm-event timescale" Le Mesnil et al study the influence of karst on runoff generation processes using a large number storm event a large number of catchments with variable degree of karst coverage. Event descriptors are compared to catchment attributes and climatic descriptors. Despite large variations among their catchments, they show that karst areas show increased infiltration from rivers during floods, increased flood times with lower peaks, and lateral losses to other catchments.

A previous version of the manuscript got a general positive feedback but recommendations in the frame of major revision had to be performs, mostly in terms of more detailed analysis especially yon the interpretations of the interbasin groundwater flow (IGF). Following the remarks of the AE and the referees, the authors performed substantial revisions and provided a strongly improved version of their manuscript:

- Subsection 5.3 was extended with an additional figure and elaboration about the relationship between IGF and fast and slow flow components

- Added more information and a new figure in subsection 5.1 that provides more insights into the seasonality of IGF in different regions.

- Also added information and a new figure in subsection 5.2 that visualizes the differences between topographic and subsurface, "hydrogeologically active" catchment areas.

- Elaborated the novelty of this study, added more justification for chosen methods and added more complete literature review, clarified the meaning of some of the used variables, and clarified in the conclusions that their work addresses gaps of karst research at the stream and river scale (not the aquifer scale).

For all those reasons, I feel confident recommending publication.

We are grateful for this positive feedback on the work done for manuscript revision.